# The mechanical response of talin

Mingxi Yao[1,*], Benjamin T. Goult[2,*], Benjamin Klapholz[3,4], Xian Hu[1], Christopher P. Toseland[2], Yingjian Guo[1], Peiwen Cong[1], Michael P. Sheetz[1,5] & Jie Yan[1,6,7]

Talin, a force-bearing cytoplasmic adapter essential for integrin-mediated cell adhesion, links the actin cytoskeleton to integrin-based cell–extracellular matrix adhesions at the plasma membrane. Its C-terminal rod domain, which contains 13 helical bundles, plays important roles in mechanosensing during cell adhesion and spreading. However, how the structural stability and transition kinetics of the 13 helical bundles of talin are utilized in the diverse talin-dependent mechanosensing processes remains poorly understood. Here we report the force-dependent unfolding and refolding kinetics of all talin rod domains. Using experimentally determined kinetics parameters, we determined the dynamics of force fluctuation during stretching of talin under physiologically relevant pulling speeds and experimentally measured extension fluctuation trajectories. Our results reveal that force-dependent stochastic unfolding and refolding of talin rod domains make talin a very effective force buffer that sets a physiological force range of only a few pNs in the talin-mediated force transmission pathway.

[1] Mechanobiology Institute, National University of Singapore, Singapore 117411, Singapore. [2] School of Biosciences, University of Kent, Canterbury CT2 7NJ, UK. [3] Wellcome Trust/Cancer Research UK Gurdon Institute, University of Cambridge, Tennis Court Road, Cambridge CB2 1QN, UK. [4] Department of Physiology, Development and Neuroscience, University of Cambridge, Anatomy Building, Downing street, Cambridge CB2 3DY, UK. [5] Department of Biological Sciences, Columbia University, New York, New York 10027, USA. [6] Department of Physics, National University of Singapore, Singapore 117542, Singapore. [7] Centre for Bioimaging Sciences, National University of Singapore, Singapore 117546, Singapore. * These authors contributed equally to this work. Correspondence and requests for materials should be addressed to M.P.S. (email: ms2001@columbia.edu) or to J.Y. (email: phyyj@nus.edu.sg).

Cellular mechanical forces at cell adhesions are emerging as a critical factor governing adhesion growth, maturation and cell migration[1]. In the last decade, many mechanosensing proteins have been identified to be important for cell spreading, migration and development. In recent years, emerging evidence has shown that talin acts as a mechanosensor, converting applied physiological forces generated from actomyosin contraction to cellular responses such as adhesion growth and maturation[2–6].

Talin, an adhesion plaque protein, links the integrin-mediated cell–matrix contacts to the actin cytoskeleton. When mechanical cues arise, either internally such as myosin-based contractility or externally in the cases of matrix stretching, talin acts as a key component of the force-transmission pathway that propagates these mechanical perturbations between cytoskeleton and cell adhesions. This has been shown to regulate diverse physiological and pathogenic processes, including embryonic development and heart homeostasis, as well as cancer metastasis[7–9].

The mechanosensing functions of talin rely on the capability of conformational changes in the 13 talin rod domains (Fig. 1a) under force that change the interactions between talin and other cellular factors[2–4]. The most established mechanosensitive function of talin is its interaction with vinculin, a scaffold protein that engages and remodels the local F-actin network, strengthening the adhesion linkage[10–12]. It has been revealed that force-dependent unfolding of the first 3 talin rod domains, R1–R3, drastically increases talin binding to vinculin by force-induced exposure of five cryptic vinculin-binding sites (VBS)[4,6]. However, the mechanical response of the domains that comprise the full-length talin rod (FL-talin rod) has not been studied previously.

The classical view of talin has it making the force-bearing linkage between the extracellular matrix (ECM) and the actomyosin contractile machinery[13], bound to integrin via the F3 domain in the N-terminal talin head and to actin via a number of actin-binding sites located along the talin rod, including one at the very C terminus in R13 (ref. 14; Fig. 1b). In this scenario, F3 and R13 form attachment points, and forces are exerted on talin linearly across the molecule from the 80 amino-acid flexible neck domain to the domains of the talin rod (Fig. 1b). Therefore, the domains in this force transmission pathway, R1–R12, directly experience tensile forces and consequently have the potential to undergo physiologically relevant conformational changes in response to force.

It has been shown by single-molecule super-resolution imaging that the end-to-end distance of talin in living cells undergoes rapid fluctuations at focal adhesions[15]. The dynamics of this fluctuation correlate with actin retrograde flow and actomyosin contraction. In a single stretching phase in the fluctuation, the end-to-end distance of talin can extend rapidly from 50 to ~350 nm, over a timescale of ~10 s. As the folded length of talin is ~80 nm, this suggests that some of the talin rod domains must be unfolded by force *in vivo*[15]. Yet information about the fluctuation of force in talin during such extension fluctuations remains unknown. This information is crucial, not only for

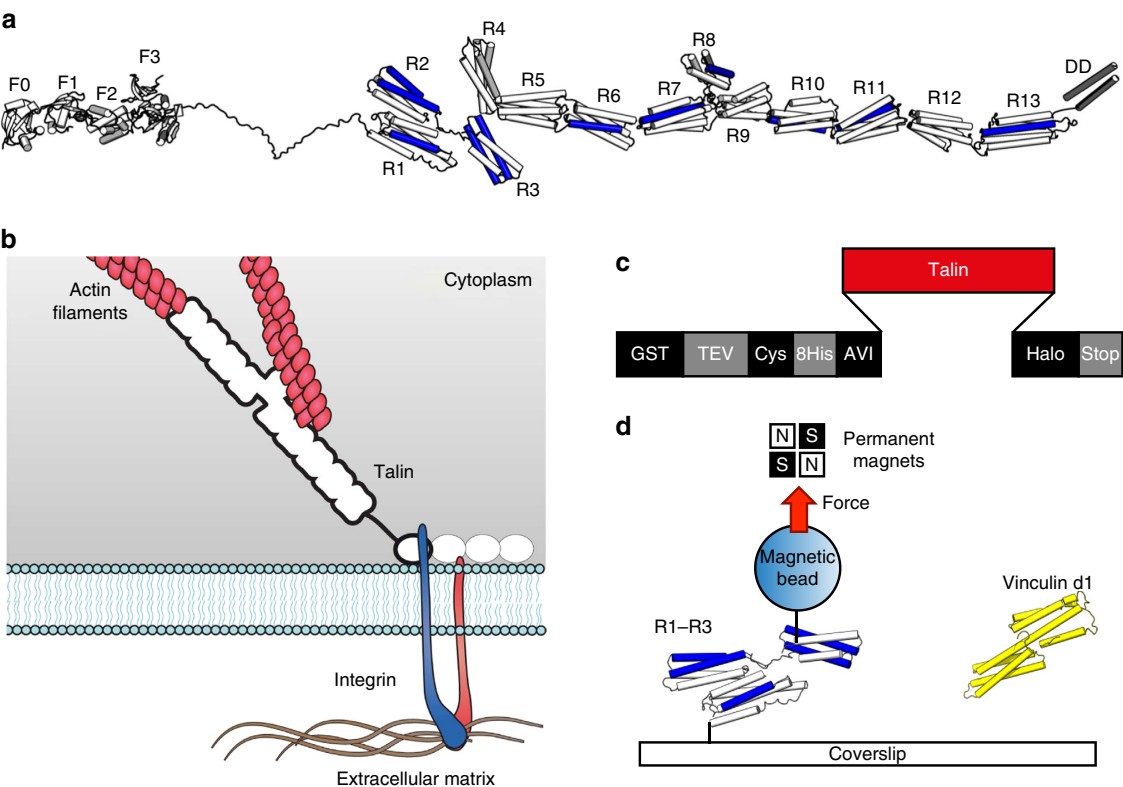

**Figure 1 | Stretching talin.** (**a**) Structural model of FL-talin. The head domain comprising F0–F3 is separated from the 13 rod domains (R1–R13) via an unstructured 80 residue linker. The last helix is a dimerization domain (DD). The 11 VBS are shown in blue. (**b**) The classical view of talin's function, linking the ECM:integrin complex to the actin cytoskeleton. In this scenario force is exerted across the talin domains outlined in bold. (**c,d**) Experimental set-up. (**c**) The custom 'stretch vector' used in these experiments. Talin fragments (red) were subcloned into a multiple-cloning site and expressed to produce a protein with a glutathione S-Transferase-tobacco etch virus (GST-TEV) site for rapid purification leaving an N-terminal Avi-tag and C-terminal Halo-tag for stretching. (**d**) The fragment of interest (R1–R3, for example) was specifically tethered between a glass coverslip and a 2.8-μm diameter paramagnetic bead using Halo-tag and Avi-tag/streptavidin chemistry. Binding partner of interest (vinculin d1, for example) can be added in solution to investigate its force-dependent interaction with talin.

understanding force transmission along the integrin-mediated pathway from ECM to cytoskeleton but also for deciphering how talin serves as a mechanosensor via mechanosensitive interactions between the talin rod and its numerous binding partners.

In this study, we systematically investigated the mechanical stabilities of the FL-talin rod comprising all 13 rod domains (R1–R13) using magnetic tweezers. Further, constructs containing subsets of talin domains were investigated to pinpoint the characteristic mechanical responses of different regions of the talin rod. By measuring the force-dependent unfolding/refolding rates of talin, and correlating this with the experimentally measured *in vivo* talin extension trajectories[15], we were able to show that the average level of force acting on talin in focal adhesions during extension fluctuation is in the range of 5–10 pN. Talin, via its interaction with vinculin, has been thought of as a force-transducing molecular clutch[16]. Our results reveal that talin additionally serves as a force buffer during large strain change, a property conferred by its multiple rod domains. Finally, we define the force range in the talin-mediated force transmission pathway in living cells and the relevant force range over which mechanosensitive interactions can take place.

## Results

**Mechanical response of the talin rod.** Talin is comprised of 18 structured domains; F0, F1, F2 and F3, which make up the atypical FERM domain in the talin head[17,18] coupled via an 80 amino-acid unstructured linker to 13 helical bundles[19], R1–R13, that make up the talin rod[12]. To explore the mechanical response of talin we sought to study all 13-rod domains, (R1–R13, residues 482–2482), that make up the FL-talin rod (Fig. 1a,b and Supplementary Fig. 1). To simplify the cloning and production of stretchable fragments we developed a 'stretch vector' containing a multiple-cloning site (Fig. 1c).

To study the mechanical properties of the FL-talin rod, a force-cycle procedure was carried out. In the beginning of each force cycle, the force applied to the protein increases linearly from 0.5 to ~40 pN at a constant loading rate of ~3.8 pN s$^{-1}$. After reaching 40 pN, at which point all domains are unfolded, the applied force was quickly reduced to ~0.5 pN for 60 s to allow the unfolded domains to refold. By repeating such cycles tens of times on each molecule for more than five independent tethers, several hundreds of unfolding events were obtained, which gave the statistics of unfolding forces (Supplementary Fig. 2).

Figure 2a shows the force-extension curve of the FL-talin rod. The unfolding is remarkably quantized in nature, and discrete stepwise extension was observed. The step sizes (~30–40 nm) correspond to the unfolding of protein domains of ~120–170 amino acids (50–70 nm contour length; Fig. 2b lower panel and Supplementary Fig. 2), consistent with the size of talin bundles.

Remarkably, the unfolded domains rapidly refold as evidenced by near-identical unfolding responses on multiple extension cycles obtained from the same tether. The talin rod is composed of 62 amphipathic α-helices, and despite this repetitive nature, the helices refold very specifically into their native domains. This is in contrast to titin, where sequential domains require low conservation to prevent misfolding[20]. As talin is likely to experience multiple extension–retraction cycles in a cellular context, this ability to rapidly refold back to its native state is likely to be critical for it to function as a mechanosensor.

In our previous studies on R1–R3, R3 was unique in undergoing equilibrium unfolding at ~5 pN (ref. 6). Only the first unfolding event in Fig. 2a resembles R3 (Fig. 2a, inset). Therefore, we conclude that the 5 pN unfolding signal is from R3 confirming R3 as the weakest domain, and consequently the initial mechanosensor, in talin.

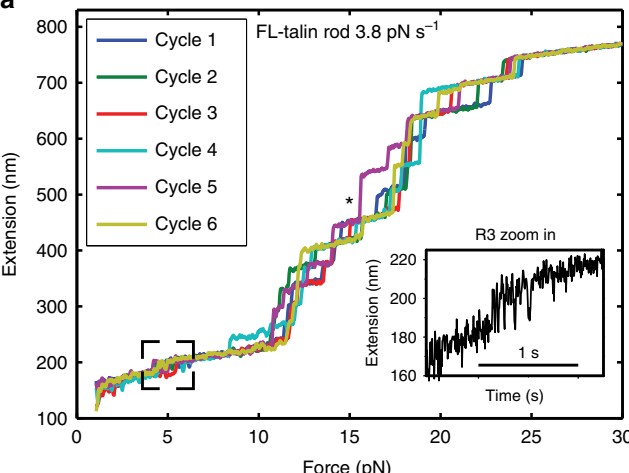

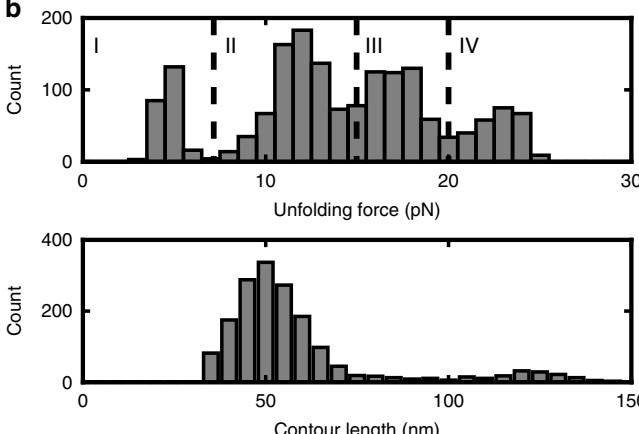

**Figure 2 | Mechanical properties of talin. (a)** Unfolding force-extension curves of FL-talin rod at the loading rate of 3.8 pN s$^{-1}$. The different curves represent repeated force cycles from a single protein tether (inset: the R3 unfolding step occurs at 5 pN; scale bar, 1 s). The data were smoothed by 0.05 s time window for clearer presentation. **(b)** The unfolding force histogram and corresponding unfolded contour length of FL-talin rod constructs (data from three independent tethers) under constant loading rates of 3.8 pN. The unfolding signals are grouped into four groups I–IV based on the distinctive unfolding forces.

The unfolding force distribution of the FL-talin rod in Fig. 2b can be divided into four distinct groups: group I (< 8 pN); group II (8–15 pN); group III (15–21 pN); and group IV (> 21 pN) separated by boundaries at ~8, ~15 and ~21 pN, respectively, at the loading rate of 3.8 pN s$^{-1}$. We note here that there is an ~10% uncertainty in force calibration (see Methods); therefore, domains close to the boundary can be assigned to either one of the adjacent groups. The talin domains contributing to the unfolding forces within each group have similar mechanical stabilities. Group I has an unfolding peak located at ~5 pN, which can be attributed to the unfolding of R3 (ref. 6). The large number of events in this group is due to the near-equilibrium nature of R3 unfolding under our loading rate that results in multiple counts of unfolding signals in one cycle of stretching. The rest of the talin rod domains contribute to the unfolding forces in groups II–IV.

**R8 is protected from force via insertion in R7.** The FL-talin rod contains 13 domains; however, only 12 unfolding steps are

observed in Fig. 2a. Interestingly, one of the unfolding steps has an extraordinarily large step size, ~80 nm (* in Fig. 2a), nearly double the step size of the other unfolding steps, suggesting two domains might unfold cooperatively. Knowledge of the domain structure of talin rod identified R7–R8 as being atypical, with R8 inserted into the loop between two helices of R7 (refs 12,21,22; Fig. 3a). This unusual topology led to the hypothesis that R8 might be protected from force, as it lies outside the force transmission pathway along talin[21]. This domain arrangement also implies that unfolding of R8 can only occur after R7 is unfolded. Such cooperative unfolding could explain the atypically large, 80-nm stepwise increase in extension observed at ~15–20 pN (Fig. 2a). Consistent with this hypothesis, a single step of ~80 nm was observed at ~15 pN when the construct R7–R8 was stretched (Fig. 3b). Unfolding of R8 alone occurred at a force of ~5 pN (Fig. 3c). Therefore, the unfolding of R7 at 15 pN results in immediate unfolding of R8.

We also stretched a segment containing R7–R9 construct, and observed two unfolding steps: a large (~80 nm) unfolding step (Fig. 3d), followed by a smaller (~40 nm) step. Both R7–R8 and R9 in the construct R7–R9 belong to group III in Fig. 2b.

**R4–R6 and R9–R12 domains unfold at 10–20 pN.** To further dissect the mechanical properties of talin, we studied the force responses of the other talin rod domains. Figure 4a,b shows the force-extension curves of R4–R6 and R9–R12 regions of talin rod, respectively. The domain boundaries were designed based on our previous structural studies on the FL-talin rod[12]. As expected the R4–R6 and R9–R12 regions have three and four distinct unfolding steps in the force range of 10–25 pN, respectively, indicating all helical bundles in these segments of talin are in the force transmission pathway and unfold individually. Together with our previous studies on R1–R3 and the data obtained for FL-talin rod (Fig. 2a) and R7–R8 (Fig. 3b), these results provide a comprehensive understanding of the mechanical stability of the FL-talin rod and its subregions.

Comparing with the unfolding force distribution obtained from FL-talin rod (Fig. 2b), two domains in R4–R6 belong to group II and one belongs to group III. In R9–R12, two domains are in group II, one (R9) in group III and one in group IV. Applying this analysis to R1–R3 construct show R1–R2 of talin has one domain in group III and one in group IV (ref. 6; Fig. 4c), meaning we have characterized the mechanical stability of talin R1–R12 into the four groups.

**Unfolding and refolding kinetics of rod domains under force.** When talin is stretched in cells, its two termini are physically coupled to the integrin tail via F3, and the actin cytoskeleton via R13 (Fig. 1b). The end-to-end distance of talin acts as an external constraint governing the conformational states of talin. We therefore sought to simulate the talin structural states and force in the talin-mediated force transmission pathway during its extension fluctuations *in vivo*. To do so, knowledge of the force-dependent unfolding/refolding rates of talin domains is needed. This information can be obtained from the force-dependent unfolding/refolding kinetics measured in experiments.

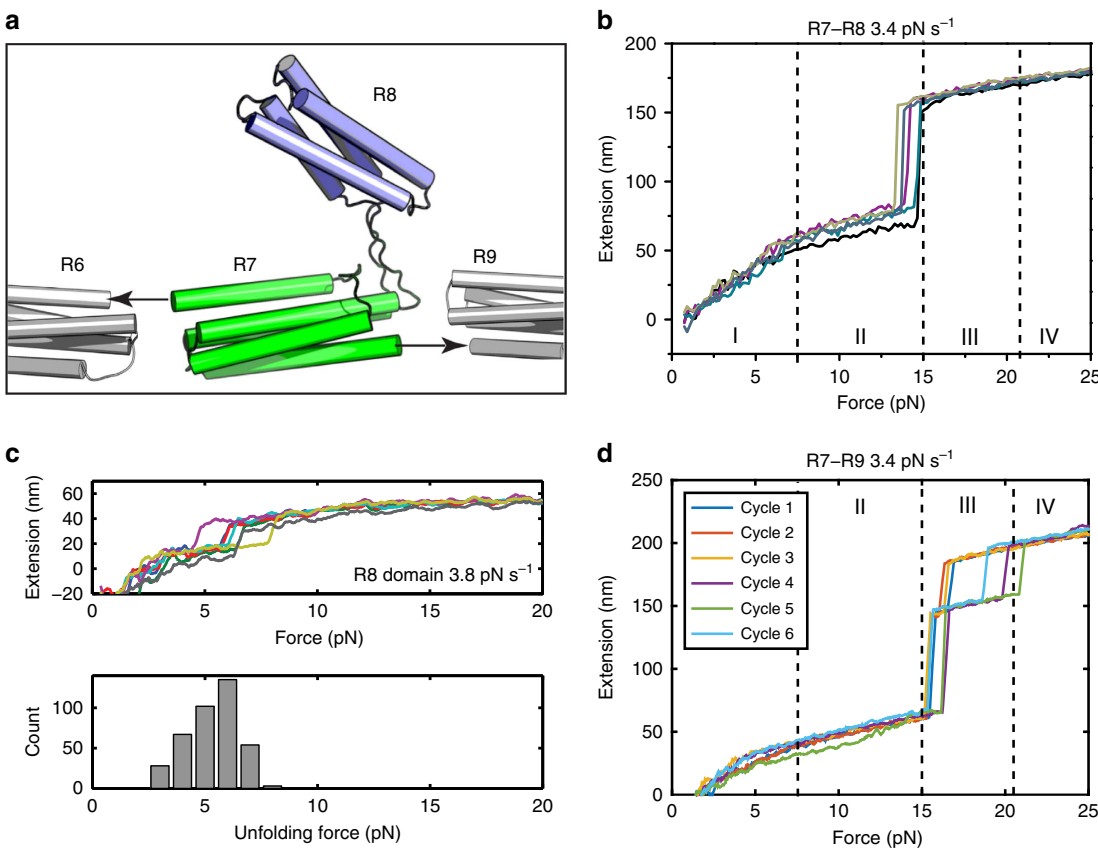

**Figure 3 | Mechanical properties of the talin R8 domain.** (**a**) Structure of R7–R8 showing the unique domain organization with R8 (blue) inserted into a loop in R7 (green). The arrows show the direction of the neighbouring domains R6 and R9 and the applied force. (**b**) Unfolding force-extension curves of R7–R8 at 3.4 pN s$^{-1}$. (**c**) Unfolding force-extension curve and unfolding force histogram of R8 alone at 3.8 pN s$^{-1}$. (**d**) Unfolding force-extension curves of R7–R9 construct at 3.4 pN s$^{-1}$ loading rate. The data were smoothed by 0.05 s time window for clearer presentation. In **b** and **d**, the unfolding force ranges corresponding to respective groups defined in Fig. 2b are indicated.

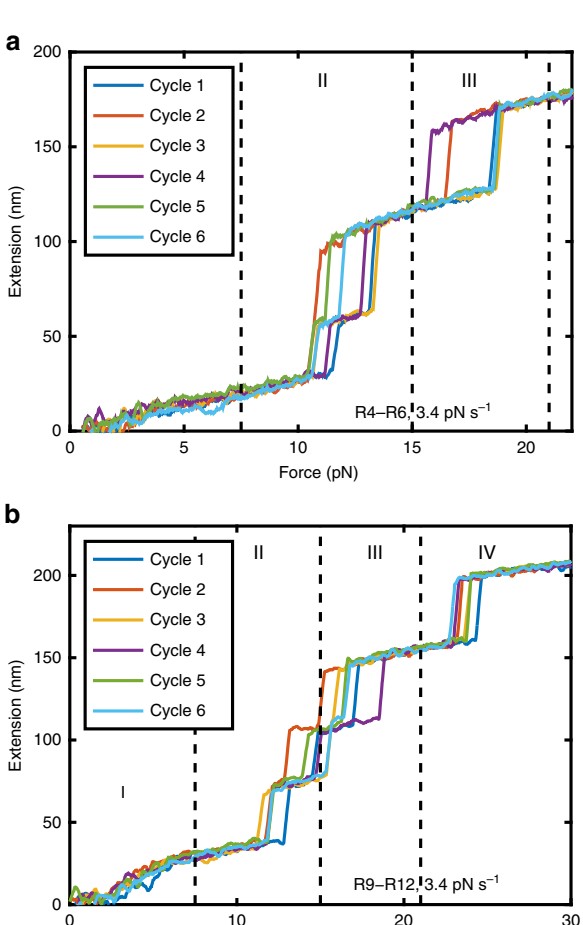

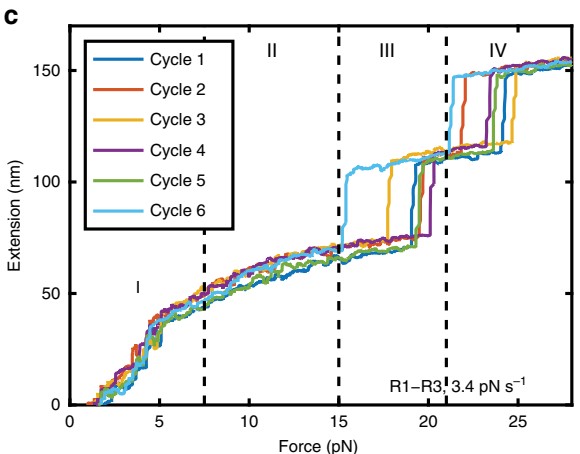

**Figure 4 | Mechanical properties of talin sub-segments. (a,b)** Unfolding force-extension curves of (**a**) R4–R6, (**b**) R9–R12 and (**c**) R1–R3 constructs at 3.4 pN s$^{-1}$ loading rate. The data were smoothed by 0.05 s time window for clarity of presentation. The unfolding force ranges corresponding to respective groups defined in Fig. 2b are indicated.

As shown in Fig. 2b, the unfolding forces of talin rod domains can be divided into four groups based on similarity in their mechanical stabilities. Therefore, we expect that the unfolding kinetics of domains in each group can be described by a single set of kinetics parameters. Unfolding events corresponding to the groups II–IV in Fig. 2b were non-equilibrium one-way transitions. The transition distance in mechanical unfolding of proteins can be approximated as constants; therefore, Bell's

model was applied to fit the data[23] (Methods: Unfolding kinetics parameters).

The unfolding kinetics parameters, $\Delta x^i_{u,0}$ (transition distance) and $k^i_{u,0}$ (unfolding rates at zero force) of groups $i = \text{II–IV}$ obtained by best fitting to the unfolding force histograms obtained at two loading rates of 3.8 and 0.4 pN s$^{-1}$ (Fig. 5a) are provided in columns II–IV in Table 1. Because R3 undergoes near-equilibrium transitions, its kinetics parameters were directly determined by constant force experiments (Fig. 5b,c and Methods: Unfolding/folding rates of R3 domain) and are provided in column I in Table 1. $\Delta x^i_{u,0}$ of the talin domains are estimated in the range of 3–5 nm.

The force-dependent folding rates for the talin rod domains were obtained by a sequential force jump procedure (Fig. 5d,e, Methods: Folding kinetics parameters). Figure 5d shows the average number of domains refolded as a function of holding time at several forces from 1 to 5 pN for talin R9–R12. Assuming the refolding of the individual helix bundles is independent of each other, the average number of folded domains over a holding time interval $\Delta t$ is as follows: $N(\Delta t) = \sum_{i=9}^{12} p^i(\Delta t)$. Here $p^i(\Delta t) = 1 - \exp(-k^i_f(F)\Delta t)$ is the folding probability of a particular domain over the time interval and $k^i_f(F)$ is the corresponding rate of refolding at the force $F$.

For talin R7–R8 and R9–R12, we found that a single exponential factor ($k_f$) was sufficient to fit the data at each force (Fig. 5d, solid lines), suggesting that the four domains in R9–R12 have similar refolding rates. In contrast, R1–R2 requires two exponential factors for good fitting. Similarly, data obtained from R4–R6 need at least two rate constants for good fitting (Supplementary Fig. 3) (Methods: Folding kinetics parameters). These results suggest that domains in these regions of talin rod have heterogeneous folding kinetics. Regarding R3, its refolding rates were directly measured by analysing the lifetime distribution of the unfolding states under constant force equilibrium measurements (Fig. 5c and Methods: Unfolding/folding rates of R3 domain).

The refolding transition involves a transition distance that is the extension difference between the transition state and the flexible peptide chain of the unfolded state. The flexible peptide leads to a force-dependent transition distance; therefore, the Bell's model that assumes a constant transition distance can no longer be applied. As such, the more general Arrhenius relation was used to extrapolate the force-dependent folding rates to a broader force range[23] (Methods: Folding kinetics parameters). These unfolding/refolding parameters allowed us to estimate the force-dependent folding rates outside the force range in which the experiments were conducted.

**Force in talin-mediated force transmission pathway.** *In vivo*, the extension of talin fluctuates in the range of 50–350 nm due to stochastic catch and release from actin retrograde flow[15,24]. As the folded talin rod has a contour length of only ∼80 nm (ref. 25), extension fluctuation over such a wide range indicates that some domains in the talin rod must be in unfolded states (Fig. 6a). An important question is how such stochastic unfolding and refolding of individual talin rod domains during *in vivo* extension fluctuation affects force in the talin-mediated force transmission pathway.

To estimate the force in the talin-mediated force transmission pathway, we performed stochastic kinetics simulations using the Gillespie algorithm[26] based on the force-dependent unfolding and refolding rates of talin rod domains (R1–R12) (Tables 1 and 2 and Supplementary Table 1, Methods: Kinetics simulations). R13 was not included in this simulation because it forms part of the actin-binding interface and unfolding of the domain could lead to tether detachment.

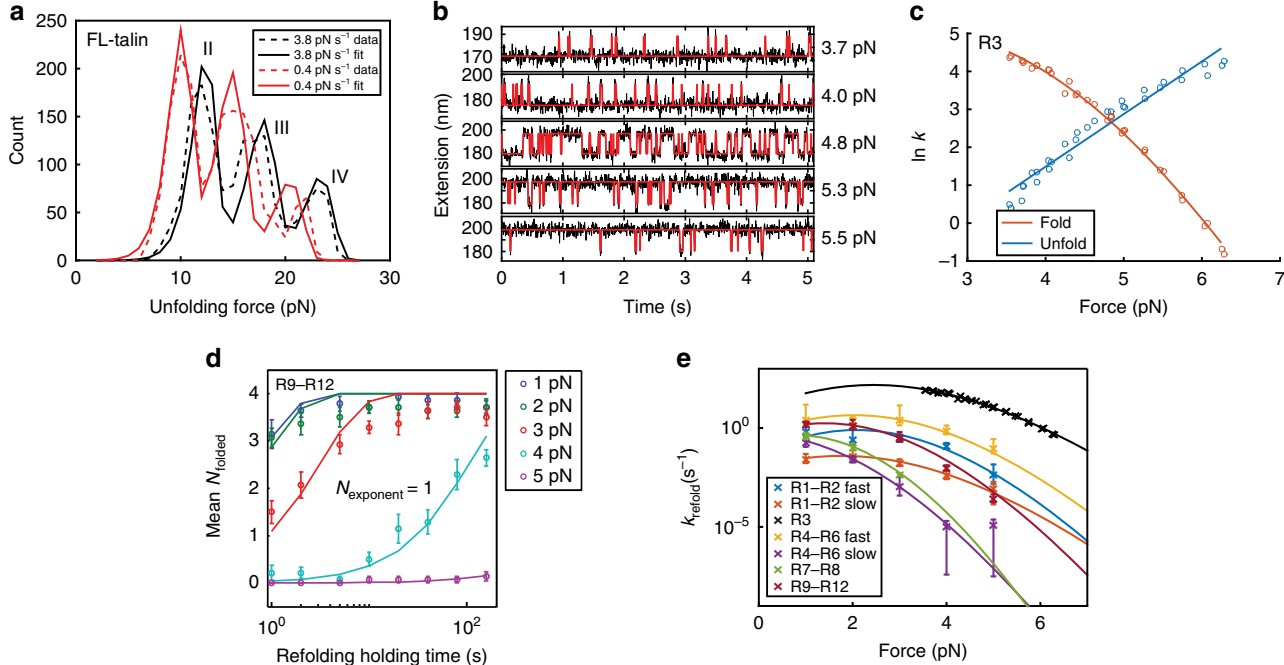

**Figure 5 | Quantification of force-dependent talin unfolding/refolding kinetics.** (**a**) Fitting the unfolding force histograms of group II–IV FL-talin rod domains using the Bell's model at two different loading rates of 3.8 (black) and 0.4 pN s$^{-1}$ (red). The dashed lines represent experimental data and solid lines represent the least squared sum fitting of the experiment data using Bell's model. (**b**) The equilibrium unfolding/refolding fluctuation of talin R3 domain at constant forces. The black curve denotes the raw data and the red curve denotes the digitized two-state fluctuations determined by hidden Markov model[6]. (**c**) The force-dependent unfolding and refolding rates of talin R3 domain determined by analysis of the lifetime distributions of unfolded/folded states under constant forces[6,38]. The dots represent rates determined from experiments. The red solid line denotes the fitting curve of the refolding rates based on Arrhenius law. The blue solid line denotes the fitting curve of the unfolding rates based on Bell's model. (**d**) The mean number of refolded domains in R9–R12 after an initially completely unfolded R9–R12 by high force was held at low forces (1–5 pN) for varying time intervals. The refolding rate at each force can be determined by exponential fitting to the data obtained at corresponding force (solid lines). The error bar denotes standard error of the mean. (**e**) The force-dependent folding rates of talin sub-segment constructs. The error bars denote 95% confidence interval of folding rate estimation. The solid lines denote fitting based on Arrhenius law. For talin R4–R6 and R1–R2, two exponents were required to account for the different refolding kinetics of the individual domains (denoted by 'fast' and 'slow') (Supplementary Fig. 3). For talin R9–R12, the refolding kinetics can be described by a single exponent. The refolding rate of R3 (black crosses) and its fitting with Arrhenius law (black solid curve) was determined from constant force measurements (the same data in **c**, plotted for comparison).

**Table 1 | Unfolding kinetics of talin domain groups described by Bell's model.**

| Domain groups | I | II | III | IV |
|---|---|---|---|---|
| Counts | 1 | 4 | 5 | 2 |
| $\Delta x_{u,0}$ (nm) | 5.7 ± 0.3 | 4.1 ± 0.1 | 3.1 ± 0.1 | 3.4 ± 0.1 |
| $k_{u,0}$ (s$^{-1}$) | 0.018 ± 0.006 | 2.5 ± 0.68 × 10$^{-5}$ | 4.2 ± 1.1 × 10$^{-6}$ | 1.7 ± 1.2 × 10$^{-8}$ |

The mean and variance of Bell's fitting parameters by bootstrapping of the talin-unfolding data (computed using bootstrap function of MATLAB with 1,000 resample number).

Figure 6b shows an example of the evolution of force (top panel) and the number of unfolded domains in FL-talin rod (middle panel). In this case the talin extension was extended from an initial 70 nm where all rod domains are folded to 250 nm over 200 s, followed by holding at 250 nm for an additional 1,800 s (bottom panel). The middle panel shows that the number of unfolded domains increases rapidly during the extending process, and then fluctuates around five whilst the extension is held at 250 nm. Correspondingly, force also increased rapidly during the extending phase in a saw-tooth pattern caused by unfolding of domains resulting in abrupt force decreases. Apparently, such unfolding events can prevent force accumulation during talin elongation. When the molecule was held at 250 nm extension, force fluctuated in a narrow range of 5.6 ± 0.6 pN.

Figure 6c shows the average force (blue, left axis) and the average number of unfolded talin rod domains (red, right axis) as

a function of extension. The average number of unfolded domains increases as the extension increases, while the force remains at an average plateau of <10 pN even when the talin rod was stretched to a long extension of ~400 nm where most domains are unfolded.

We then applied these simulations to experimentally measured extension trajectories of single talin molecules in living cells[26] (Fig. 6d,e, panels in the third row). Figure 6d,e shows the simulated results when talin fluctuated at a shorter extension (~100 nm; Fig. 6d) and a longer extension (~200 nm; Fig. 6e), respectively. As expected, significantly more domains were unfolded when talin fluctuated at longer extension, exposing more cryptic VBS (Fig. 6d,e, panels in the second row). In contrast, in both cases the force remained at a similar level of ~5–6 pN (top panels). These results indicate that force-dependent stochastic unfolding and refolding of talin rod

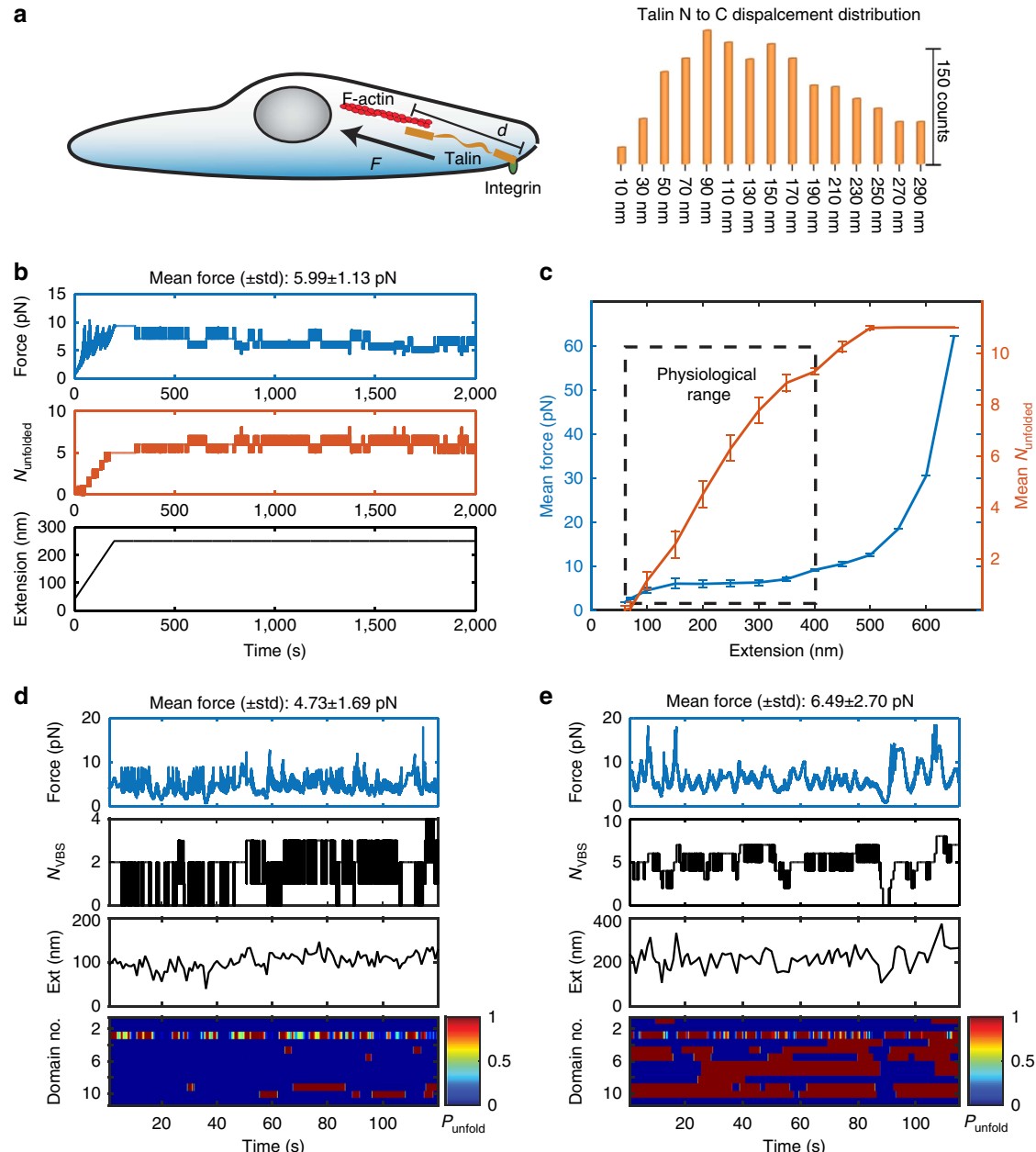

**Figure 6 | Simulation of force in talin-mediated force transmission pathway.** (**a**) Illustration of the *in vivo* talin extension (left panel) and the measured distribution of talin extension in fibroblast cells[15]. (**b**) Simulation of the force (top panel) and the number of unfolded domains (middle panel) during changing the FL-talin extension as indicated in bottom panel. (**c**) Average force (data in blue) and number of unfolded domains (data in red) in FL-talin rod as a function of extension. Solid connecting lines are provided for visual guiding. The error bars denote s.d.'s. The black box denotes the physiological range of extension measured in fibroblast cells as shown in **a**. (**d**,**e**) Simulated force fluctuations (top panels) in FL-talin based on two different levels of FL-talin extension fluctuations around (**d**) 100 and (**e**) 200 nm recorded from living cells[15]. Talin end-to-end fluctuations measured experimentally from *in vivo* single-molecule localization studies[15] in fibroblasts are shown in the third panels. First panels: the estimated force fluctuation on the talin rod. Second panels: $N_{VBS}$ denotes the number of exposed VBS. Bottom panels: heat maps showing the unfolding probability of each individual talin rod domain during the time evolution. The talin was significantly more extended in **e** than in **d**.

**Table 2 | Refolding kinetics of talin domains described by the Arrhenius law.**

| Domain no. | 1–2 | | 3 | 4–6 | | 7–8 | 9–12 |
|---|---|---|---|---|---|---|---|
| Counts | 1 | 1 | 1 | 2 | 1 | 1 | 4 |
| $L_0$ (nm) | 18.2 ± 0.7 | 12.6 ± 0.9 | 15.5 ± 0.1 | 15.7 ± 1.9 | 4.4 ± 1.9 | 13.3 ± 1.1 | 14.5 ± 1.0 |
| $k_{f,0}$ (s$^{-1}$) | 0.11 ± 0.08 | 0.019 ± 0.009 | 22.2 ± 2.5 | 1.0 ± 0.8 | 0.46 ± 0.3 | 0.39 ± 0.14 | 0.93 ± 0.3 |

The mean and variance of Arrhenius fitting parameters for 1,000-time bootstrapped talin-folding data by resampling the refolding data rates from a *t*-distribution for each force 1,000 times and fitting each set of resampled rates to Arrhenius law.

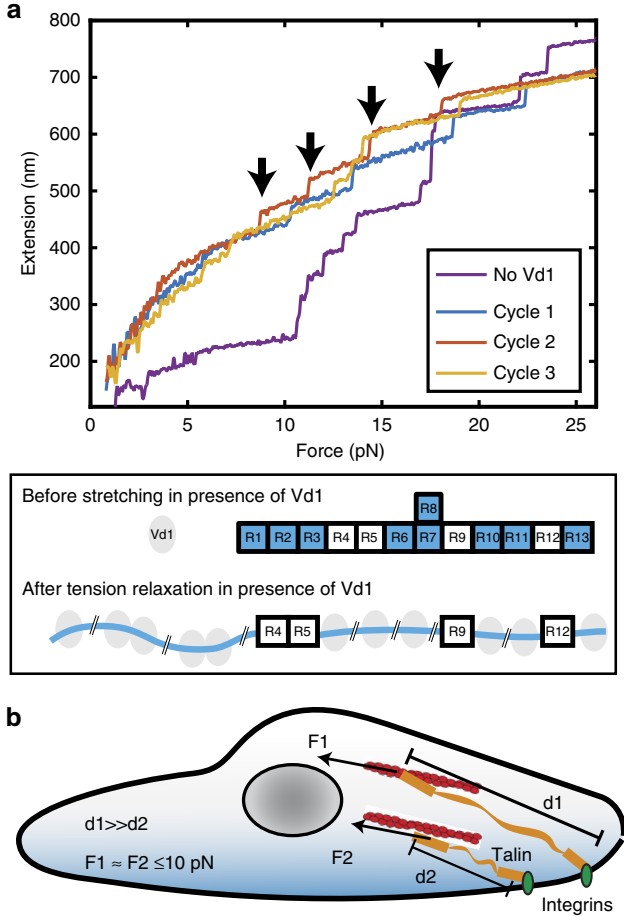

**Figure 7 | The effect of Vd1 on the FL-talin rod. (a)** The purple curve denotes the force-extension curve of FL-talin rod in the absence of Vd1. The other curves (cycles 1–3) depict three representative unfolding force extension curves in the presence of 100 nM Vd1. The four unfolding steps in cycle 2 are marked by arrows. The data were smoothed by 0.05 s time window for clearer presentation. Bottom: cartoon showing the changes in conformation of talin rod in the presence of Vd1. **(b)** Talin as a force buffer in the cellular force transmission pathway. Stochastic unfolding/refolding of talin rod domains in response to changes in force ensures that force across the whole talin-mediated, force-transmission pathway can be maintained at a low state (<10 pN) even across very different talin end-to-end extension fluctuations.

domains make talin a very effective force buffer during large extension fluctuations that sets a physiological force range of only a few pN in the talin-mediated force transmission pathway.

**Vinculin binding to FL-talin.** The results above show that stochastic talin domain unfolding during extension increase buffers the force at <10 pN. Of the 62 helices that make up the talin rod, 11 have been shown to interact with the cytoskeletal protein vinculin with nanomolar affinity at a peptide level[10]. These VBS are usually cryptic, hidden within the hydrophobic core of the folded talin rod domains[12,27]. Our previous studies of force-dependent binding of the vinculin D1 domain (Vd1) to mechanically stretched talin R1–R3 (ref. 6) revealed that mechanical unfolding of these bundles was necessary to promote binding of vinculin. However, such studies have yet to be done for the other bundles in the talin rod.

As shown in Fig. 7a, a FL-talin rod tether without Vd1 in solution showed 12 unfolding steps. After the buffer was changed to include 100 nM Vd1, we first unfolded all talin domains and reduced the force to 6 pN for 20 s, which prevents bundle refolding while permitting Vd1 binding, and then further reduced force to <1 pN for 1 min to allow unbound bundles to refold. In a following stretching phase, only four unfolding steps were observed, indicating that nine bundles could not refold at <1 pN, due to Vd1 binding when they were unfolded in the preceding force cycle. Considering only four bundles do not contain VBS (Figs 1a and 7a; R4, R5, R9 and R12), this experiment shows, for the first time, that all the domains that contain VBS in talin are able to bind vinculin over physiological force range, and vinculin binding prevents domain refolding once force is released. Furthermore, as we observe 12 unfolding steps the first time we stretch FL-talin rod in the presence of Vd1 our findings show that all 11 of the VBS in talin are cryptic in the absence of force.

**Discussion**

In this work, by stretching individual talin molecules we have defined the mechanical properties of talin, which, in conjunction with structural knowledge, enable us to model the *in vivo* force response of talin. Our results reveal that during the early stages of contractile cell spreading, even over a wide range of talin extensions, the average force in each talin monomer does not exceed 10 pN. Thus, talin sets the physiological force range that defines the mechanical stability of cell–matrix adhesions.

Previous studies have shown that the extension of talin rod is fluctuating in a range of 100–300 nm in living cells[15], but how the force fluctuates during dynamic talin stretching is unknown. In addition, recent single-molecule fluorescent resonance energy transfer (smFRET) tension sensor measurements report 7–10 pN of average force in talin[28,29], but the cause of this force in talin is unclear. Our results provide answers to these unknowns by suggesting that stochastic unfolding of talin domains, when talin is stretched over a physiological relevant extension range, maintains the average force in the whole talin-mediated, force-transmission pathway at below 10 pN (Fig. 7b). The ability of talin to act as a force buffer may be important for the maintenance of adhesion integrity. Future experiments should be directed to understand the lifetime of the linkages in this force range, to provide insights to the catch and release kinetics in the force-transmission pathway.

Talin's mechanosensing functions rely on its force-dependent interactions with its binding partners, which include integrin, actin, vinculin, RIAM, DLC-1, α-synemin and so on[13]. Some ligands bind to folded talin domains while vinculin is known to bind to cryptic motifs in folded bundles. Force-induced talin domain unfolding will release binding partners that bind to folded talin and stimulate binding of vinculin[12], triggering mechanosensing signals. As discussed above, the unfolding, refolding kinetics of talin sets a level of a few pN on average, which can transiently reach >10 pN during force fluctuations. This creates an interesting question of how talin's binding partners are affected by such force fluctuations.

Our results reveal important insights into the mechanics of talin within a focal adhesion. When the talin C-terminal actin-binding site captures the retrograde actin flow, the end-to-end distance of talin can extend up to several hundreds of nanometres without a significant increase in force. However, the number of activated VBS increases as length extends, increasing vinculin binding. Our results show that the vinculin head can bind to all mechanically unfolded domains that contain VBS, and does not dissociate after force is released.

Consistently, previous *in vivo* talin extension measurements showed over-expression of vinculin head caused an increased extension of talin ($>300$ nm) with reduced fluctuation. Since the bound vinculin head prevents talin domain refolding, such binding should further relax the force in the talin-mediated force transmission pathway. However, the impact on force by full-length vinculin binding could be more complex, as vinculin engaging additional F-actin filaments might lead to increase in force[30]. Vinculin autoinhibition will augment this interplay further as the force-dependent binding constant will result in force regimes where vinculin autoinhibition is favoured over VBS binding. Such competing interactions with different force dependences will be interesting continuations of this work.

As well as exposing VBS, domain unfolding also disrupts binding sites for ligands that bind to the folded forms. This is the case for RIAM binding to the high-affinity RIAM-binding site in R2–R3 (ref. 12). This suggests that force may switch the binding partner from one to another, activating different cellular signalling. This also potentially adds a temporal element to force-dependent binding; domains that have a high mechanical stability can be unfolded at high force, yet remain unfolded at much lower forces. Our force-dependent folding rate measurements show that if talin is maintained above 5 pN then many of the domains will not refold over minutes of timescale. Further, it adds a spatial component since smaller nascent adhesions, where the forces are lower, may never exceed these mechanosensing force thresholds. Talin has a complex pattern of binding partners with overlapping binding sites[13], and changes in the availability of binding sites may help determine what binds when and where.

Our results show that the talin rod domains have a variety of mechanical stabilities, as seen from the clustered unfolding forces in Fig. 2b. Such discrete mechanical responses may be an advantage by providing a graded mechanosensing. *In vivo*, talin is extended by the actin retrograde flow, and the resulting force built up inside the talin rod drives structural transitions of the bundles. In the course of the talin extension process, the weaker bundles unfold first (at low extensions) and the stronger ones unfold later (at long extensions). Such gradual unfolding leading to disruption of binding sites on folded domains, combined with the sequential exposure of VBS, may allow fine-grained mechanosensing of cells over a wide range of environmental rigidities.

The unfolding transition distance of talin domains was determined to be in the 3–4 nm range, which is considerably larger than many previously studied protein domains, such as titin I27 and filamin immunoglobulin (Ig) domains, unfolded at a similar loading rate[23,31]. A longer transition distance implies that the unfolding forces of talin domains are less sensitive to loading rates compared with domains with shorter unfolding transition distances. This feature of rod domains gives talin the robustness of buffering force in the force-transmission pathway during talin extension fluctuation in living cells.

It is striking that R9, which forms the high-affinity auto-inhibitory domain that regulates talin activity by binding to the F3 domain[32,33] is mechanically stable (Fig. 3d), remaining folded at forces $>15$ pN even in the presence of vinculin. This suggests that, even in the presence of tensile stress, R9 will remain intact and thus able to fine-tune adhesion dynamics. This autoinhibitory effect regulates adhesion dynamics; disruption of autoinhibition in *Drosophila* causes defects in morphogenesis due to increased adhesion stability resulting from reduced turnover of talin within adhesions[7].

Among the 13 domains of the talin rod, R3 and R8 are unique in terms of their structural features. They are both four-helix bundles that contain a threonine belt that destabilizes their

hydrophobic core[12]. The four-helix unzipping geometry and weak hydrophobic core imply that they are mechanically unstable[22] and we now show that this is the case. Our previous study confirmed that R3 is mechanically weak, unfolding at $\sim 5$ pN forces. By stretching the FL-talin rod, we show that R3 is the weakest talin domain, most likely to unfold first when talin is under force.

Although R8 shares a similar domain architecture to R3, its positioning within talin is markedly different, excluded from the force transmission pathway by insertion in the R7 domain. Consequently, R8 is protected, and only unfolds cooperatively with the more mechanically stable R7. R8 is a binding hotspot containing multiple binding sites for a number of 'LD-motif' containing signalling molecules such as RIAM, DLC1 and α-synemin[13], binding to its folded structure. It also forms part of the actin-binding site (ABS2)[34]. The mechanical protection provided by R7 is essential for R8 function as a signalling hub, enabling it to remain folded at forces significantly greater than it could itself withstand. R8 also contains a VBS that will only be exposed in response to high forces, and it is tempting to speculate that under such conditions, binding of vinculin prevents refolding and silences this signalling hub. The R7–R8 topology is a striking example of how a binding hub like R8 can be positioned within a molecule so as to remain folded at forces that would normally cause it to unfold.

One important point that requires consideration when studying talin as a mechanosensor is the emerging complexity of talin's role in forming the connection between the integrin:ECM complex and the actomyosin machinery. The classical view has been that talin binds integrins at its N terminus and F-actin at its C terminus. In this arrangement the forces exerted on talin are recreated well in this current study. As shown in Fig. 1b, there are additional actin-binding sites[34] and there is also a second integrin-binding site[35]. Recent work has revealed alternative mechanisms for talin to mediate integrin function, forming alternative conformations relative to the membrane[22,36], suggesting talin may also sense different force vectors, which will be interesting to explore further.

The work described here couples structural information with mechanobiology to allow the precise mechanical response of a complex mechanosensor to be understood. The quantitative description of the structural states of the talin rod under force established by this study serves as a basis to explore the different scenarios that exist at cell–matrix adhesions.

## Methods

**Protein expression.** All talin fragment plasmids (Supplementary Fig. 1) were synthesised by PCR using mouse *talin1* cDNA template and cloned into a custom expression vector (Fig. 1c). Proteins were expressed in *Escherichia coli* BL21(DE3) cultured in Luria-Bertani (LB) media[14]. The GST-tagged constructs were purified using glutathione Sepharose resin (GE Healthcare) and eluted by TEV cleavage.

**Single-molecule manipulation.** The single-molecule manipulation experiments were carried out using a custom high-force magnetic tweezers platform that can exert forces up to 100 pN with $\sim 1$ nm extension resolution for stuck bead at 200 Hz sampling rate[37].

For given magnets and bead, the force is solely dependent on the magnet-bead distance $F(d)$, which can be calibrated based on a method described in our previous publication, which has an $\sim 10\%$ uncertainty due to the heterogeneous bead sizes[37]. On the basis of the calibrated $F(d)$, multiple ways of force control were achieved by changing $d$ with time accordingly. A constant force is achieved when a constant $d$ is maintained. For loading rate control where force increases linearly with time, $F(t) = r \times t$, the magnet-bead distance is programmed to change with time as $d(t) = F^{-1}(r \times t)$, where $F^{-1}$ is the inverse function of $F(d)$ and $r$ is the loading rate.

For the unfolding experiments, the protein of interest was immobilized on the glass coverslip of a laminar flow chamber and to a 3-μm paramagnetic bead (Dynabeads M270 streptavidin) using Halo-tag/Halo-ligand and biotin/streptavidin chemistry[6,38] (Fig. 1d). Briefly, glass coverslip was cleaned in an

ultrasonic cleaner in 10% Decon 90 solution, followed by acetone and isopropanol for 30 min each. Then the coverslips was silanized by 1% APTES (Sigma-Aldrich) in methanol for 20 min and rinsed clean by methanol. The APTES-coated coverslip was assembled into a flow channel and $NH_2$-O4-Halotag ligand (Promega) was immobilized on to the coverslip through glutaraldehyde (Sigma-Aldrich) crosslinking. The channel was blocked by 1 M Tris (pH 7.4) for 30 min followed by 1% BSA in $1 \times$ PBS and 0.1% Tween-20 over night. Protein of interest containing Halo-tag and biotinylated Avi-tag was immobilized by flowing $\sim 0.1\,\mu g\,ml^{-1}$ protein into the channel for 20 min. And then streptavidin-coated M270 beads were added to the channel to form the tether.

For the refolding rate measurement, a 576-bp DNA linker was incorporated between the protein and the magnetic bead. This reduced the potential effects of steric hindrance of the magnetic beads on the protein-refolding rates. In this case, a 576-bp DNA from lambda phage vector was amplified by PCR with a Thio-labelled 5′ primer and biotin-labelled 5′ primer. The 576-bp DNA was covalently immobilized to epoxy-activated 3 μm paramagnetic beads following the manufacturer's instructions (Dynabeads M270-epoxy). The concentration of DNA during incubation was kept low ($\sim 0.01\,ng\,\mu l^{-1}$) to minimize multiple binding on a single bead. During the experiments, the talin construct of interest was immobilized to a glass coverslip using halo-tag chemistry. The buffer was then switched to one containing $0.02\,mg\,ml^{-1}$ streptavidin for 20 min followed by incubation with the DNA-coated paramagnetic beads. The multivalent streptavidin acted as a bridge that linked the DNA handle to talin. All unfolding and refolding experiments were carried out in $1 \times$ PBS, 1% BSA , 1 mM dithiothreitol and 0.1% Tween-20.

**Unfolding kinetics parameters.** The unfolding kinetics parameters of talin rod domains were determined by fitting the force histogram of FL talin rod to the unfolding force probabilities of Bell's model at two loading rates (0.4 and $3.8\,pN\,s^{-1}$). The R3 domain typically underwent near-equilibrium unfolding/refolding transition at $\sim$ 5 pN in our experimental loading rates (group I in Fig. 2b), whose kinetics parameters were obtained separately by constant force measurements as described in the main text. The unfolding kinetics parameters for the remaining talin rod domains were obtained by fitting to unfolding force histogram based on the Bell' model at forces above 7.5 pN, as described below.

The resulting unfolding force histograms with R3 unfolding excluded (Fig. 5a) were globally fitted by the formula $\sum_i N^i \rho^i_{k^i_{u,0}, \Delta x^i_u}(F)$, where $\rho^i_{k^i_{u,0}, \Delta x^i_u}(F) = \frac{k^i_{u,0}}{r}\exp\{\frac{\Delta x^i_u F}{k_B T} + \frac{k_B T k^i_{u,0}}{\Delta x^i_u r}[1 - \exp(\frac{\Delta x^i_u F}{k_B T})]\}$ is the unfolding force probability density distribution of the group $i = II$, III or IV predicted based on the Bell's model[39]. $r$ is the force loading rate used in the experiments, $k^i_{u,0}$ is the unfolding rate at zero force, $\Delta x^i_u$ is the transition distance from the folded state to the transition state and $N^i$ is the number of unfolding events in each group. The unfolding force data were resampled 1,000 times (bootstrap function of MATLAB) and for each subsample the kinetics parameters were fitted using lsqcurvefit solver of MATLAB. The mean and s.d. of the 1,000 parameters were shown in Table 1.

**Folding kinetics parameters.** The force-dependent folding rates for the talin rod domains were obtained by a sequential force jump procedure. During each experiment, a talin subdomain was fully unfolded by increasing force to 30 pN at $3.8\,pN\,s^{-1}$. Following the unfolding step, the force was jumped to a lower force between 1 and 5 pN, and the tether was held at the force for various time intervals. The number of domains that refolded during the time interval $\Delta t$ was indicated by the number of unfolding events in the subsequent unfolding procedure. This procedure was carried out > 20 times for each specified force and folding time interval to obtain the average and standard errors of the number of domain folding. Note that a 576-bp DNA handle was inserted between the C terminus of each protein construct and the magnetic bead using streptavidin bridge to prevent bead-surface interaction at low forces that might affect the refolding kinetics.

Assuming the refolding of the individual helix bundles are independent of each other, the average number of folded domains over a time interval $\Delta t$ is as follows: $N(\Delta t) = \sum^{12}_{i=9} p^i(\Delta t)$. Here $p^i(\Delta t) = 1 - \exp(-k^i_f(F)\Delta t)$ is the folding probability of a particular domain over the time interval and $k^i_f(F)$ is the corresponding rate of refolding at the force $F$.

The highly flexible peptide chain in the unfolded state leads to a force-dependent folding transition distance; as such, the Bell's model is no longer applicable. Therefore, the more general Arrhenius law is used to extrapolate the force-dependent folding rates to a broader force range[23]: $k_f(F) = k^0_f exp(\int^F_0 \Delta x_f(F')dF')$. Here, $\Delta x_f(F) = x_{TS}(F) - x_{unfolded}(F)$ is the force-dependent refolding transition distance that can be calculated based on the force-extension curves of the transition state $x_{TS}(F)$ and unfolded peptide chain $x_{unfolded}(F)$.

The transition state, which is likely a partially folded state or a chain of rigid helices, is approximated as a rigid body of certain size $L_0$ and therefore $x_{TS}(F) = L_0 \times \coth(\frac{F \times L_0}{k_B T}) - \frac{k_B T}{f}$ (ref. 23). Most single-chain α-helices are not stable in solution; therefore, we approximated the unfolded state as a randomly coiled peptide chain, with its force-extension curve described by the worm-like-chain model using Marko–Siggia formula with persistence of 0.8 pN (ref. 40). Treating $L_0$ and $k^0_f$ as fitting parameters, the experimentally measured force-dependent refolding rates were fitted as described above (Fig. 5e). To obtain statistically robust

fitting parameters, we generated a set of 100 rate values at each force based on the t-distribution, with the mean of determined rate value and scaling parameter obtained from the 95% confidence interval obtained from rate fitting (fitexp function of MATLAB). Then the rates were randomly resampled to generate 1,000 sets of force-dependent rates for fitting to the Arrhenius relation ($k_f(F) = k^0_f \exp(\int^F \Delta x_f(F')dF')$). The average and s.d.'s of the 1,000 refolding kinetics parameters were presented in Table 2.

**Unfolding/folding rates of R3 domain.** The force-dependent unfolding/folding rates of R3 domain were determined from constant force measurements of unfolding/refolding time lapse. As shown in Fig. 5b, talin R1–R3 constructs were held at several different forces around 5 pN, and 30 min of talin unfolding/folding dynamics were recorded for each force. The unfolding/folding fluctuation data at each force were assigned to unfolded and folded states by fitting to a Hidden Markov model described in our previous publication[38]. Then histograms of lifetimes were generated for the unfolded and folded state and fitted to an exponential decay function (fitexp function of MATLAB) to obtain the folding/unfolding rates at each force (Fig. 5c). To obtain kinetics parameters for R3, the force-dependent unfolding rates were fitted directly to the Bell's model: $k_u(F) = k^0_u \exp(\frac{\Delta x^i_u F}{k_B T})$ and the force-dependent folding rates were fitted using the Arrhenius relation described in the above section.

**Kinetics simulations.** Unlike optical tweezers or atomic force microscope (AFM) that control the extension of molecules and measure force fluctuation, magnetic tweezers constrain the force applied to the molecule and measure the extension fluctuation. Therefore, they do not directly measure force fluctuation in the talin-mediated force transmission pathway. However, this information can be obtained through simulations based on the force-dependent unfolding and refolding rates measured by magnetic tweezers.

Kinetics simulation of the force evolution of talin as a function of extension fluctuations was carried out by an in-house written MATLAB programme using Gillespie algorithm[26]. Briefly, the talin rod domains are treated as a one-dimensional lattice model. At a given extension, the force in the rod is estimated based on the structural states (that is, folded or unfolded) and force-extension curves of the domains in the corresponding states. In response to the force, each domain undergoes stochastic structural transitions based on its force-dependent unfolding and refolding transition rates. Both the time to the next transition event and the domain involved in that transition are stochastically determined based on the classic Gillespie algorithm[26]. After each transition, the structural states of the lattice were updated, and the resulting force was calculated. Iteration of this process results in evolution of the structural states, causing force fluctuation in the talin rod. Using the transition rates listed in Tables 1 and 2 and Supplementary Table 1, the simulation can reproduce the unfolding force histogram observed in our experiments (Supplementary Fig. 4)

**Data availability.** The MATLAB codes of the kinetics simulation is provided in Supplementary Data 1. The data that support the findings of this study are available from the corresponding authors on request.

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

## Acknowledgements

We thank David Critchley for stimulating discussions and critical reading of the manuscript. We also thank the protein expression facility of the Mechanobiology Institute for protein purification. Y.J. is funded by the National Research Foundation, Prime Minister's Office, Singapore under its NRF Investigatorship Programme (NRF Investigatorship Award No. NRF-NRFI2016-03) and grants from the National Research Foundation through the Mechanobiology Institute Singapore. B.T.G. is funded by BBSRC grant (BB/N007336/1) and B.K. by BBSRC grant (BB/L006669/1). C.P.T. is funded by MRC grant (MR/M02060/1). M.P.S. is funded by National Research Foundation through the Mechanobiology Institute Singapore and NIH grant GM113022.

## Author contributions

J.Y. and M.P.S. designed the research and supervised the experiments; M.Y., X.H. and Y.G. performed the experiments; B.T.G., B.K. and C.P.T. contributed new reagents and constructs; M.Y., B.T.G., J.Y. and P.C. interpreted and analysed the data; M.Y., B.T.G., J.Y. and M.P.S. wrote the paper.
