## [Peer review file · Nature Communications]

Reviewers' Comments:

Reviewer #1 (Remarks to the Author)

In this manuscript Yao et al. examine the force-dependent unfolding and refolding of the talin rod domain. To do so, they use a magnetic tweezers apparatus that allows them to stably exert low forces over long periods of time. They also examine the unfolding and refolding of subdomains within the talin rod. Their principal observations are that i) unfolding forces (at a given pulling rate) are clustered at roughly 5, 12, 17, and 22 pN, ii) all of the talin subdomains are capable of fully refolding on biologically relevant timescales, iii) subdomain 3 in particular unfolds at ~ 5 pN, iv) the resulting mixture of the various unfolding/refolding forces results in a roughly constant force of ~ 5 pN for talin extensions of between 100 and 200 pN, and the exposure of 2-5 vinculin binding sites in this force range. The later result comes from Monte Carlo simulations performed using force-dependent unfolding and refolding parameters determined experimentally. The authors conclude that "force-dependent stochastic unfolding and refolding of talin rod domains make talin a very effective force buffer that sets a physiological tension range of only a few pN in the talin-mediated force transmission pathway."

Summary recommendation: This is a potentially excellent paper, though a few important details need to be addressed before it is ready for publication. Assuming these can be successfully accomplished, this paper should definitely be published in Nature Communications.

Detailed comments:

1) The fit parameters in Table 1 are presented without statistical uncertainties. Given the complexity of the models being used it is essential that the fit uncertainties be determined. I would recommend the bootstrap or a conceptually similar approach, since the fit errors derived from the nonlinear least squares algorithm implemented by Matlab can greatly underestimate the uncertainties resulting from finite sample size.

2) Although I leave it to the discretion of the Editor, I recommend the source code for the Monte Carlo simulation be included as supplemental information. This is simple to implement, and will allow others in the field to directly assess this component of the study.

3) It is somewhat puzzling that the authors do not use their magnetic tweezers setup to directly test the prediction of the Monte Carlo algorithm, namely that talin buffers force over a large range in extensions. One of the virtues of the magnetic tweezers apparatus is that it makes pulling experiments that last multiple minutes easy to implement. Given the experimental prowess of the authors I would guess that this experiment would be straightforward, and would greatly strengthen (or obviate) the principal conclusion of the paper, as stated in the quote from the abstract above. However, I would not insist on this experiment, provided that the authors can demonstrate that my assumption of its feasibility is incorrect, since I don't want to hold up publication too long. In this later case, however, the authors should highlight this limitation the study in the text so that non-experts are aware of it.

4) Optionally, I suggest that the authors streamline and clarify their discussion of the interplay of talin unfolding and vinculin recruitment as a function of load. There are a lot of good points, but this material could be better organized. I was also left wondering about how the arrangement of type II-IV domains might impact force transmission to integrins. Is there any rationale for their arrangement within the rod?

Reviewer #2 (Remarks to the Author)

This is an interesting manuscript that describes the unfolding and refolding of the talin rod domain in response to application of force. The effect of force on talin has been well studied, and the idea that talin unfolds in response to force is well accepted. This study extends the previous studies by focusing on all 13 helical bundles within the rod domain. The authors present evidence that unfolding and rapidly refolding of talin in response to force is random, thereby allowing a low level of tension to be maintained across the molecule. While these findings will likely be of interest to the talin community, they do not represent a significant advancement over what is already known. Furthermore, it remains speculative that talin is a "force buffer" as the investigators purport. Hence the work is perceived as being an incremental advancement rather than one which will have a sustained impact on the field or one that will garner widespread readership.

Reviewer #3 (Remarks to the Author)

In their manuscript, entitled "The mechanical response of talin", Yao et al. present experimental evidence that in addition to its scaffold function in focal adhesions, talin acts as a force buffer by maintaining tension in a physiological force range. This was found to be conferred by talin's C-terminal rod domain with its 13 helix bundle sub domains, of which force-dependent stochastic unfolding and refolding have been studied. Force ranges for this process as well as kinetic rates were experimentally determined by means of single-molecule force spectroscopy in magnetic tweezers.

The presented force buffering mechanism was found to be in agreement with previous findings, that talin, which spans about 80 nm in its folded state, can rapidly undergo extension changes between 50 and 350 nm. In addition, it was shown, that all predicted cryptic vinculin binding sites in the talin rod domain could bind vinculin in mechanically stressed talin.

Yao et al. utilized a magnetic tweezer setup to study talin rod domain unfolding. Structural dissection of the entire rod domain into constructs of 2-4 helix bundles allowed classification into four different unfolding force regime groups between 5 and 25 pN. Shielding effects of mechanically less robust subdomains like R8, which is excluded from the force propagation path by intact R7, were also elucidated.

Whilst certain aspects of talin subdomain mechanical stability and its influence on binding or dissociation of signalling and mechanotransduction molecules have been studied previously by Sheetz, Yan and others, this work gives an extended and more conclusive understanding of talin's role as a force sensor and, as has now been established, force buffer. Single-molecule in vitro data on stochastic unfolding and refolding of subdomains in talin's rod domain was additionally correlated to in vivo data by simulating tension fluctuations in the force transmission pathway.

Magnetic tweezers are well suited to study this biomolecular system as they enable high resolution in the probed force range. The chosen construct design and the site specific immobilization strategy warrant control over the tether assembly in the measurement setup and thus unambiguity of the obtained data. The collected data and its presentation is clear and justifies the drawn conclusions.

The authors use Arrhenius and Bell type models to fit unfolding forces and rates. While this general theoretical framework is well established, there are some inconsistencies and/or points needing clarification in regard to the kinetic models, as outlined in the suggestions below.

The conclusions drawn by the authors can be considered reliable for the given model of talin rod domain behavior under mechanical stress. The data was obtained with single talin molecules, isolated from the cellular context and even more so from their various biomolecular interactions (in addition to vinculin, which was actually accounted for in one of the experiments). The findings are still valid and of relevance even in a more expanded molecular network as found in focal adhesion. The simulation of tension fluctuations in the talin mediated force transmission pathway in vivo extends the model derived from the single-molecule study. Further, limitations of the studied geometry and conclusions derived thereof are addressed, with respect to scenarios where the force does not propagate through the full talin molecule in a linear way. This takes into account that

talin for example harbors further attachment sites for actin along the talin rod, while the experimental setup allowed only for simulations of actin attachment in the C-terminal R13 subdomain. Also, implications of the presented results with respect to the autoinhibitory interaction between talin domains R9 and F3 and thus its ability to associate with integrin are addressed.

Both presentation and discussion of the results as well as the general correlation of the drawn conclusions to previous findings and the current understanding of talin's role as a force sensor in focal adhesions are very clear and conclusive.

I recommend publication of the manuscript by Yao et al. in Nature Communications, after the following issues have been thoroughly addressed:

1. p.2: Concerning the extension range talin can span, different values are mentioned (here 50 nm to >300 nm, compare p. 6 and 8), for consistency I would recommend giving the maximum reported range.

2. p. 2, p. 8 and p.10: How does interaction of the folded helix bundles with other proteins/protein domains (e.g. actin and RIAM) potentially interfere with the observed unfolding behavior? Aside from the discussed displacement of binding partners from initially folded domains upon their unfolding, can stabilizing effects and thus a shift in unfolding force to higher values for the respective rod sub domain be expected? Could the authors please comment on this and briefly discuss this in the manuscript?

Especially in case of the mechanically least stable R3 domain that interacts with RIAM, would the authors expect that the presence of RIAM in the buffer during the MT unfolding experiment have a significant effect on the unfolding barrier observed for R3? If this was feasible, would it in fact be possible to add a measurement in presence of RIAM or the respective RIAM domain construct that confers binding to (R2/)R3?

3. p.3 (and online methods): Could the authors please explain or give a reference for how the constant loading rate was achieved in the experimental setup? Magnetic tweezers usually operate in the constant force regime, as e.g. outline in reference 35 of the manuscript.

4. p. 3 and Fig. 2/ Fig. S2: Is there any hierarchy in the sequence of the helix bundle domains and their respective mechanical stability? I.e. are domains closer to the N-terminus more like to unfold at lower forces (corresponding to groups II and III) than the C-terminal ones (the R9-R12 construct appears to harbor the bundle with the highest mechanical stability)?

To add to completeness and conclusiveness the actual force-extension traces for the R1-R3 should be given as well, even though the histograms are included in Fig. S2. Are these data from this study or the previous ones on R1-R3 unfolding (ref. 4 and 6) and if so, could you please clarify this in the manuscript?

Could the authors please comment on reasons for the different yields (counts) in observed unfolding events for the different constructs (R4-R6, R7-R9 and R9-R12, Fig. S2)?

In Fig. S2b the low unfolding contour length found for the R1-R3 construct does not appear to be observed in the CL histogram of the FL construct. Is that due to the scale or is the isolated domain construct in some way destabilized such that one of the domains is only partially folded leading to a smaller CL increment?

5. p.4/ Fig. 2a: What is the timescale for the inset, displaying the fluctuation corresponding to R3 un-/refolding?

6. p.5: The observed Δx values, i.e. the distances to the transition state, for unfolding of the individual helix bundles are fairly large, compared e.g. to typical distances to the transition state from AFM-based protein unfolding experiments, which tend to be ≤ 1 nm. Could this be related to the proposed function as a force buffer and the observed un-/refolding transitions? A short

discussion of this aspect in the manuscript would add to its conclusiveness.

7. p. 6 and Fig. 5/S3: Could the authors please comment on the possible relevance of the observed heterogeneous folding kinetics and shortly discuss possible implications?

8. p. 9: Should Fig. 3c in the last paragraph actually be Fig. 3d, which shows unfolding of the R9-R12 construct?

9. p. 10: Whilst the F3 domain that forms the autoinhibitory interaction with R9 was not included in the rod-only construct of talin that was probed, could the authors give an idea of what force range the unbinding between F3 and R9 would occur at? Will this interaction likely be under mechanical strain in a physiological context, e.g. by membrane contacts of the N-terminal FERM region (upstream of F3 and the integrin/R9 interaction side)? Considering the force range in which talin has been shown to confer tension compensation, would you expect that the removal of the autoinhibitory contact is also (at least partially) force driven or rather only a competition effect with integrin binding?

10. Online methods:

For the refolding kinetic studies a DNA linker was inserted between bead and protein construct to prevent steric hindrance effects by the bead. Was there reason to believe that this could happen? And why would that not interfere with the other unfolding experiments?

When utilizing DNA linkers, the surface density on the beads was kept low to prevent multiple interactions. How was that accounted for when measuring with the commercial SA-beads? Despite the observation of unfolding of all the domains included in the respective construct, was there another criteria for ensuring that only a single molecule was tethered by each bead? Along this line, could the authors briefly comment on the yield of the tethering strategy (compare p. 3: "for more than five independent tethers")?

11. Fig.3: Comparison between panels b) and d) suggests that the mechanical stability of the cooperatively unfolding domains R7 and R8 is slightly increased when R9 is also present in the construct. Is this a systematic effect or is it within the error of the determined unfolding force at the given loading rate. Considering this, could the unfolding behavior that is observed for isolated sub domain groups without a doubt be compared to their mechanical stability in full length talin? This should be shortly addressed in the manuscript.

12. Fig. 3d (and 4b): The traces suggest that in the particular case of R9 unfolding, that recurring unfolding and refolding increases the stability of R9, i.e. the unfolding force. While this is likely a coincidence, could there be any reason to believe that some of the helix bundles exhibit some kind of mechanical memory?

13. Fig. 5: There are several points needing clarification in regard to the kinetic models:

- Fig. 5c: Does $\ln(K)$ refer to $\ln(k)$ (i.e. $\ln(k_u)$ and $\ln(k_f)$, respectively)? The capitalized K would otherwise indicate the ratio of the two rates, i.e. the equilibrium binding constant.

- More importantly, the fitted curve for the folding rate is clearly not linear in the log-lin plot, which should be the case when fitting with an Arrhenius type relation. Could the authors please explain this discrepancy? In fact, it appears that the curve for R3 unfolding given in Fig. 5e has been inserted in the 5c plot! Could the authors please correct this figure and include the actual data points with the proper Arrhenius fit?

- Also, could you please discuss the deviation of the blue data points in Fig 5c from the linear fit and to what extent the approximation of fitting the data according to the Arrhenius law is still reasonable?

- Was the unfolding data in this case (Fig. 5b and c) really fitted according to the Bell model, even though the data arose from a constant force experiment? If this was the case, could the authors please motivate this approach?

- Fig. 5d: No data points for refolding rates at 4 and 5 pN for construct R7-R8 are shown and possibly are also not taken into account for the fit. Is there a reason for that or also the absence of the 4pN data in Fig. S3c?

14. Fig. S1: Were all of the shown constructs really used and mentioned for the experiments presented in the manuscript (e.g. R7-R9 Δ R8 and F3-R1)? Otherwise I would suggest to either remove constructs from the list that are not relevant for the presented results or otherwise include the data corresponding to the extra constructs.

List of Main Changes

Main changes marked in red in the main text:

1. Page 20, paragraph 2: A new paragraph was added to highlight the difference in mechanical constraints applied by different single-molecule manipulation instruments between magnetic tweezers and optical tweezers/AFM.
2. Page 16, paragraph 4: A new paragraph is added to introduce the details of how force controls such as constant force and loading rate are achieved in our experiments.
3. Page 14, paragraph 2: A new paragraph was added to discuss the implication of longer transition distance of the talin domains.
4. Page 5, paragraph 4: A new sentence was added to discuss the relative error in force determination and its effect on domain assignment.
5. Page 20, paragraph 1: A new section: "Unfolding/folding rates of R3 domain" is added to the Methods to describe how to extract kinetic parameters of the R3 domain's ~ 5 pN constant force transitions.
6. Page 18, paragraph 1 and page 19, paragraph 2: Descriptions were added to the Methods on how bootstrap statistical analysis was done for unfolding/refolding kinetics parameters.

Main changes to the figures and table:

7. Table 1 - the kinetics parameters were regenerated using bootstrap method according to the request from reviewer 1.
8. Figure 5 – The fitting curves in panel e were updated using the bootstrapped parameters.
9. Figure 6 - the simulated force fluctuations and average number of unfolded talin rod domains in panels b-e were updated using the updated kinetics parameters in table 1.
10. Figure S4 – the simulated unfolding force-extension curve and unfolding force histogram were updated using the bootstrapped parameters.
11. Figure 4 – a new panel c was added for the force-extension curve of R1-R3 domain
12. Figure 2 – time scale bar was added to the inset of Figure 2a.

Captions are updated accordingly in each involved figure and table changes.

Point-to-Point Responses to Reviewers' comments

Reply to reviewer #1:

1. The fit parameters in Table 1 are presented without statistical uncertainties. Given the complexity of the models being used it is essential that the fit uncertainties be determined. I would recommend the bootstrap or a conceptually similar approach, since the fit errors derived from the nonlinear least squares algorithm implemented by Matlab can greatly underestimate the uncertainties resulting from finite sample size.

Response:

{

We thank the reviewer for this suggestion.

For unfolding kinetics parameters, we performed 1000 bootstrap resampling (Matlab bootstrp function) based on the unfolding forces recorded at each loading rate for all domains except R3. Each bootstrap resampling yielded an estimate of the unfolding kinetics parameters. The average values and the standard deviations were obtained from the 1000 bootstrap resampling.

The bootstrap analysis for the unfolding kinetics parameters of R3 was done differently. The unfolding rate of this domain at each force was obtained by exponential fitting to the lifetime histogram with more than 5000 unfolding transitions. The unfolding rates were obtained at more than 30 different forces. These data were bootstrap-resampled for 1000 times, which yielded the average and standard deviation of the unfolding kinetic parameters. The same analysis was applied to the refolding kinetics parameters and standard deviations for R3.

For refolding kinetics parameters of talin domains except R3, the above bootstrap analysis is not applicable due to the limited number of data points (total five force values, at each force there is only one rate value obtained by fitting with time evolution of probability of refolding). Therefore, we carried out a revised bootstrap analysis as described as follows.

At each force, the single rate value obtained by fitting with time evolution of probability of refolding came with a 95% confidence interval assuming t -distribution of the rate. We regenerated a set of 100 rate values at each force based on the t -distribution, then performed 1000 bootstrap resampling of force-dependent rates for fitting to the Arrhenius relation ($k_f(F) = k_f^0 \exp(\int_0^F \Delta x_f(F') dF')$) to obtain the average and standard deviations of the refolding kinetic parameters.

The unfolding and refolding kinetics parameters in Table 1 have been updated with the values obtained using the above statistically more robust approach. The updated values only slightly differ from previous values obtained without using the bootstrap resampling and do not change the results of the simulations. We have also updated the simulated force fluctuations in Fig. 6 using the slightly changed kinetics parameters generated by bootstrap method.

}

2. Although I leave it to the discretion of the Editor, I recommend the source code for the Monte Carlo simulation be included as supplemental information. This is simple to implement, and will allow others in the field to directly assess this component of the study.

Response:

{

We have included the source code of the kinetics simulation in the supplemental information.

}

3. It is somewhat puzzling that the authors do not use their magnetic tweezers setup to directly test the prediction of the Monte Carlo algorithm, namely that talin buffers force over a large range in extensions. One of the virtues of the magnetic tweezers apparatus is that it makes pulling experiments that last multiple minutes easy to implement. Given the experimental prowess of the authors I would guess that this experiment would be straightforward, and would greatly strengthen (or obviate) the principal conclusion of the paper, as stated in the quote from the abstract above. However, I would not insist on this experiment, provided that the authors can demonstrate that my assumption of its feasibility is incorrect, since I don't want to hold up publication too long. In this later case, however, the authors should highlight this limitation the study in the text so that non-experts are aware of it.

Response:

{
Unfortunately, the experiment that the reviewer describes is not feasible using magnetic tweezers as they directly control the force applied. Therefore, they are not a suitable method to study force fluctuation. However, it is important to stress that our analysis of the ability of talin to buffer force over a large range of extensions is based on experimentally derived force-dependent transition rates of individual domains we determined by magnetic tweezers experiments. Using these rates, one can simulate force fluctuation under extension control, or extension fluctuation under force control. In order to understand the force fluctuation in talin when talin is stretched by retrograde actin flow, we performed simulations under extension control and the results provided an explanation to the 7-10 pN force range *in vivo* reported in previous experiments (Austen et al. Nature Cell Biology 2015 and Kumar et al. JCB 2016). In addition, the simulated extension fluctuation under force control using the same set of force-dependent transition rates can be directly compared with magnetic tweezers experiments. As shown in Fig. S4, the simulated histogram of the unfolding force at a loading of 3.8 pN/s agrees with that obtained experimentally in Fig. 2.

We have included several sentences to clarify this point in the 1st paragraph of **Methods: "Kinetics simulations"**.

}

4. Optionally, I suggest that the authors streamline and clarify their discussion of the interplay of talin unfolding and vinculin recruitment as a function of load. There are a lot of good points, but this material could be better organized. I was also left wondering about how the arrangement of type II-IV domains might impact force transmission to integrins. Is there any rationale for their arrangement within the rod?

Response:

{
We thank the reviewer for the suggestion; we have streamlined the first two sections of discussion based on how the load applied to talin influences the growth and maturation of focal adhesions. Regarding the question on the arrangement of group II-IV domains, we find the domains belonging to a group scatter throughout the talin rod without apparent clustering in their positions.

}

Reviewer #2

This is an interesting manuscript that describes the unfolding and refolding of the talin rod domain in response to application of force. The effect of force on talin has been well studied, and the idea that talin unfolds in response to force is well accepted. This study extends the previous studies by focusing on all 13 helical bundles within the rod domain. The authors present evidence that unfolding and rapidly refolding of talin in response to force is random, thereby allowing a low level of tension to be maintained across the molecule. While these findings will likely be of interest to the talin community, they do not represent a significant advancement over what is already known. Furthermore, it remains speculative that talin is a "force buffer" as the investigators purport. Hence the work is perceived as being an incremental advancement rather than one which will have a sustained impact on the field or one that will garner widespread readership.

Response:

{
We thank the referee for his review and note that he finds it an interesting manuscript. However, we disagree with the reviewer's opinion that our results represent incremental advancement, which are interesting only to the talin community. It is well known that talin is a crucial molecule playing a critical role in focal adhesion growth and maturation. In spite of

many studies of talin, much critical knowledge necessary to understand the talin mediated mechanosensing function is still missing. In this article we address several important aspects of talin that either have not been investigated or remain poorly understood.

Specifically, we determined the mechanical stability of talin domains and their force-dependent transitions kinetics in the whole talin rod for the first time. This allowed us to simulate the force fluctuation in talin during talin extension change, which provides a novel mechanistic explanation (i.e., the force buffer concept) to the 7-10 pN forces in talin observed *in vivo*. Further, we show for the first time that all the talin rod domains bearing vinculin binding sites are cryptic but can be activated by mechanical force, which is another significant advancement compared to the knowledge derived from R1-R3 domains in our previous studies.

The new insights from this study will shed light to how force is buffered by large force bearing proteins in other systems such as titin in muscle. The methods to determine the kinetic rates and simulations based on the rates are novel and will be of interest to single molecule biophysics field. Further, the parameters determined will be useful for theoretical biophysicist to build quantitative models of mechanosensing.

}

Reviewer #3

We thank this reviewer for his/her insightful comments and suggestions, addressing of which has enhanced the manuscript.

1. p.2: Concerning the extension range talin can span, different values are mentioned (here 50 nm to >300 nm, compare p. 6 and 8), for consistency I would recommend giving the maximum reported range.

Response:

{

We have unified the description of talin extension span in the range of 50-350 nm.

}

2. p. 2, p. 8 and p.10: How does interaction of the folded helix bundles with other proteins/protein domains (e.g. actin and RIAM) potentially interfere with the observed unfolding behavior? Aside from the discussed displacement of binding partners from initially folded domains upon their unfolding, can stabilizing effects and thus a shift in unfolding force to higher values for the respective rod sub domain be expected? Could the authors please comment on this and briefly discuss this in the manuscript? Especially in case of the mechanically least stable R3 domain that interacts with RIAM, would the authors expect that the presence of RIAM in the buffer during the MT unfolding experiment have a significant effect on the unfolding barrier observed for R3? If this was feasible, would it in fact be possible to add a measurement in presence of RIAM or the respective RIAM domain construct that confers binding to R2/R3?

Response:

{

Yes, we expect that any factor binding to a folded protein domain should increase the thermal stability of the protein. Regarding how much the binding may increase the mechanical stability in a single-molecule stretching experiment, it depends on both the strength of binding and whether the binding modifies the transition state. The binding affinity of RIAM and R3 is in the low micro-molar range; therefore its stabilizing effect on the R3 domain is expected to be small. To test whether its effect can be detected using our magnetic tweezers, we stretched R1-R3 in the presence of 70 μ M RIAM TBS1 peptide (see figure R1 below), but we were not able to detect a significant stabilising effect on the R3

unfolding threshold compared to the data obtained in the absence of RIAM (Fig. 5B in manuscript). Characterisation of the force-dependence of proteins interacting with folded domains, whilst outside the scope of this current study, will be important to fully understand talin mediated mechanosensing.

Figure R1. Unfolding/refolding transition of R3 in the presence of 70 μM RIAM TBS1

}

3. p.3 (and online methods): Could the authors please explain or give a reference for how the constant loading rate was achieved in the experimental setup? Magnetic tweezers usually operate in the constant force regime, as e.g. outline in reference 35 of the manuscript.

Response:

{

In the magnetic tweezers experiments, the force applied to a paramagnetic bead solely depends on the distance, d , between the magnets and the bead. For a given pair of magnets, at a given d , the ratio of forces applied to two different beads equals to the ratio of the maximal magnetizations of the two beads (Chen et al., *Biophys J* 100:517–23). In our experiments, we have used a testing bead "0" to calibrate the force-distance curve $F_0(d)$ based on fluctuations of the bead tethered to a long lambda DNA molecule over a wide force range up to 100 pN. Therefore, in any experiment when a different bead "1" is used, its force-distance curve is simply related to $F_0(d)$ through a scaling factor c as: $F_1(d) = c * F_0(d)$. For a short tether such as a protein, the force can be accurately determined at forces below 10 pN based on the bead fluctuation in the direction perpendicular to both force and the magnetic field (Chen et al., *Biophys J* 100:517). In this force range, the c value can be determined in experiments. The force at the higher force range can then be obtained by direct extrapolation based on $F_1(d) = c * F_0(d)$.

Since we know $F_1(d)$ in any experiment, we can implement multiple ways of force control by changing d accordingly. A constant force is achieved when we maintain a constant d . If we change d with time through a trajectory of $d(t)$, we can change force with a programmed time trajectory of $F_1(t)$. In the case of loading rate control where force should increase linearly with time, $F_1(t) = r * t$, we just need to program $d(t)$ as $d(t) = F_1^{-1}(r * t)$, where F_1^{-1} is the inverse function of $F_1(d)$, which was implemented in our LabView program.

This information is now included in the method section of the manuscript.

}

4. p. 3 and Fig. 2/ Fig. S2: Is there any hierarchy in the sequence of the helix bundle domains and their respective mechanical stability? I.e. are domains closer to the N-terminus more like to unfold at lower forces (corresponding to groups II and III) than the C-terminal ones (the R9-R12 construct appears to harbor the bundle with the highest mechanical stability)?

Response:

{
We did not observe a strong hierarchy in the mechanical stability of the domain bundles except for R3. We find the domains belonging to a group scatter throughout the talin rod without apparent clustering in their positions. For example, the two domains that have the strongest mechanical stability were located in R1-R2 and R9-R12 regions, respectively.
}

To add to completeness and conclusiveness the actual force-extension traces for the R1-R3 should be given as well, even though the histograms are included in Fig. S2. Are these data from this study or the previous ones on R1-R3 unfolding (ref. 4 and 6) and if so, could you please clarify this in the manuscript?

Response:

{
We have included the force-extension traces of R1-R3 in figure 4 panel c. These data and the corresponding unfolding force histograms for R3 were newly generated for this study, which are consistent with the data reported in our previous study for R1-R3 (Yao et al. Sci Reps 2014).
}

Could the authors please comment on reasons for the different yields (counts) in observed unfolding events for the different constructs (R4-R6, R7-R9 and R9-R12, Fig. S2)? In Fig. S2b the low unfolding contour length found for the R1-R3 construct does not appear to be observed in the CL histogram of the FL construct. Is that due to the scale or is the isolated domain construct in some way destabilized such that one of the domains is only partially folded leading to a smaller CL increment?

Response:

{
The different yields for R4-R6, R7-R9 and R9-R12 were mainly due to the different number of repetitive unfolding/refolding cycles carried out on individual tethers until tether breaking. For all constructs the statistics were carried out on more than five independent tethers and for each tether more than 20 unfolding cycles were observed.

Regarding the question on the step sizes of R3 unfolding using the R1-R3 construct and full length talin rod, we think they are actually similar to each other. The shorter step sizes observed in R1-R3 only occupies < 10 % of the total events. Although the causes of these outliers are not clear, we suspect that they might be from some low probability of partial refolding events during fast unfolding and folding transitions at ~ 5 pN forces when the bead-surface distance is close to each other, contributing to the small fraction of smaller-step unfolding transitions at lower forces. This effect will have less impact on full length talin stretching due to a much larger bead to surface separation. Importantly, these are minority events, which do not change the overall statistics of rate determination.

}

5. p.4/ Fig. 2a: What is the timescale for the inset, displaying the fluctuation corresponding to R3 un-/refolding?

Response:

{

The fluctuations occurred at a sub-second time scale. We have included the time scale bar in the revised inset in Figure 2a.

}

6. p.5: The observed Δx values, i.e. the distances to the transition state, for unfolding of the individual helix bundles are fairly large, compared e.g. to typical distances to the transition state from AFM-based protein unfolding experiments, which tend to be ≤ 1 nm. Could this be related to the proposed function as a force buffer and the observed un-/refolding transitions? A short discussion of this aspect in the manuscript would add to its conclusiveness.

Response:

{

The Δx observed for talin rod domains were in the 2-3 nm range which is considerably larger than protein domains commonly studied by AFM force spectroscopy such as titin I27 and Filamin A Ig domains. This is not because the unfolding occurs at lower forces, because in our previous studies on these Ig domains using magnetic tweezers at similar forces (Chen et al., *Biophys J* 2011, 101: 1231; Chen, H. et al. (2015). *JACS*, 3546(1)), sub-nanometer unfolding transition distances were also observed, consistent with the values obtained in previous AFM experiments at higher forces. Therefore, we think the larger Δx values may be a feature of α -helix bundles (at least for talin). A long transition distance implies that the unfolding forces of talin domains are relatively insensitive to loading rates. This can be seen from the Bell's model which predicts that the derivative of the peak unfolding force with respect to the loading rate is proportional to $1/\Delta x$. Such insensitive dependence of the unfolding force on the loading rate has an apparent advantage to buffer the force in a certain range during fluctuations of extension in talin *in vivo*.

We have included a short discussion on the potential benefit of having such a large transition distance of talin rod domains in the revised manuscript.

}

7. p. 6 and Fig. 5/S3: Could the authors please comment on the possible relevance of the observed heterogeneous folding kinetics and shortly discuss possible implications?

Response:

{

The possible implications of the observed heterogeneous refolding kinetics for different talin domains are currently unclear. It is an interesting phenomenon and certainly warrants future studies on its potential relevancy to talin's mechanosensing functions.

}

8. p. 9: Should Fig. 3c in the last paragraph actually be Fig. 3d, which shows unfolding of the R9-R12 construct?

Response:

{

We thank the reviewer for pointing out this typo in the manuscript, which has been corrected in the revised manuscript.

}

9. p. 10: Whilst the F3 domain that forms the auto-inhibitory interaction with R9 was not included in the rod-only construct of talin that was probed, could the authors give an idea of what force range the unbinding between F3 and R9 would occur at? Will this interaction likely be under mechanical strain in a physiological context, e.g. by membrane contacts of the N-terminal FERM region (upstream of F3 and the integrin/R9 interaction side)? Considering the force range in which talin has been shown to confer tension compensation, would you expect that the removal of the autoinhibitory contact is also (at least partially) force driven or rather only a competition effect with integrin binding?

Response:

{

Without direct measurement of the F3-R9 interaction, it is hard to make an estimate of the force that can disrupt the interaction. The observation that R9 remains folded at forces > 15 pN raises the intriguing possibility that it might be a stable domain for a reason, and that this reason might be to modulate adhesion dynamics in adhesions. Talin autoinhibition has been shown to play an important role in regulating adhesion dynamics and our data reveal that R9 can remain folded even in an adhesion where talin is under force.

Where talin is under tension in the adhesion, F3 domain will be far away from R9, making it difficult to imagine F3 and R9 can be associated within the same talin molecule. However, due to the high density of talin molecules in an adhesion it is possible that the neighbouring R9 might modulate adhesion turnover by the competition effect with integrin binding (and the PIP2 enriched membrane, Song et al. Cell Res. 2012) which will help to fine tune adhesion dynamics.

}

10. Online methods:

For the refolding kinetic studies, a DNA linker was inserted between bead and protein construct to prevent sterical hindrance effects by the bead. Was there reason to believe that this could happen? And why would that not interfere with the other unfolding experiments?

Response:

{

Steric hindrance was not an issue for unfolding - In the unfolding experiments, the transitions typically occurred at forces greater than 10 pN (except R3). At such large force, the perturbation from the surface – surface interaction between coverslip and the bead is negligible. In addition, at higher forces the bead is pulled further away from surface, further minimizing the influence from bead-surface interaction. Therefore, we did not use handles in these experiments to increase the tether lifetime in experiments. For example, figure R2 below shows unfolding data of R9-R12 with a 576 bp DNA handle at a loading rate of 3.8 pN/s, which is indistinguishable to that obtained without DNA handle (Fig. 4b in the manuscript)

In contrast, refolding transitions typically occurred at smaller forces, < 5 pN. At these forces, the surface-surface interaction begins to affect the kinetics. We have done refolding experiments with and without handle for R9-R12, and shown that the measured refolding rate obtained without handle is a few fold slower than that with handle.

Figure R2. Force-extension curve of R9-R12 with 576 DNA handle during unfolding at 3.8 pN/s.

}

When utilizing DNA linkers, the surface density on the beads was kept low to prevent multiple interactions. How was that accounted for when measuring with the commercial SA-beads?

Despite the observation of unfolding of all the domains included in the respective construct, was there another criteria for ensuring that only a single molecule was tethered by each bead? Along this line, could the authors briefly comment on the yield of the tethering strategy (compare p. 3: "for more than five independent tethers")?

Response:

{

When commercial SA-beads were used, the surface density of talin is kept low by dilution. In addition, multiply linked tethers can be easily distinguished from single tethers by their characteristic force responses - unfolding of a singly formed tether only results in extension change along the force direction, while unfolding a doubly or multiply formed tether causes bead rotation due to torque rebalance, which results in displacement of the bead centroid in the focal plane. This has been discussed and demonstrated with details in our previous publication (Chen et al., *Biophys J* 2011, 101: 1231). Fig. S2 in that publication is shown below as an example.

Figure S2: Typical signal of multi-tether (left panel) and single tether (right panel). For multi-tether, when unfolding event occurs, the extension increases stepwisely. At the same time, location of the tethered bead in focal plane dx and dy jump too, which indicates the rotation of the bead due to un-balanced torque. While for single tether, dx, dy keep at constant values when protein unfolds.

Figure R3 (Fig. S2 in our previous publication (Chen, H et al. (2013). *Scientific Reports*, 3, 1642). Distinguish singly linked tethers from multiply linked tethers based on the transverse positional change during unfolding.

The yield of our tethering strategy is around one specific tether per few 100x objective observation view and in general 1-2 tethers were studied in a single flow channel. The yield is dependent on channel and protein preparation.
}

11. Fig.3: Comparison between panels b) and d) suggests that the mechanical stability of the cooperatively unfolding domains R7 and R8 is slightly increased when R9 is also present in the construct. Is this a systematic effect or is it within the error of the determined unfolding force at the given loading rate. Considering this, could the unfolding behaviour that is observed for isolated sub domain groups without a doubt be compared to their mechanical stability in full-length talin? This should be shortly addressed in the manuscript.

Response:

{
Our force calibration has a 10% uncertainty due to the variations in bead size. The details of the force calibration method and the uncertainty can be found in our previous publication (Chen et al., *Biophysical Journal*, 100, 2011). The R7-R8 was assigned in group III in the force-extension curve obtained from unfolding of full-length talin. In the force-extension curve obtained from unfolding of R7-R9, they still belong to the group III. Another example is R3, which is the other domain that can be identified into a group in the force-extension curve obtained from full-length talin, belongs to the same group I in both full-length talin and R1-R3 stretching experiments. These results suggest that there is no significant difference in the mechanical stability between domains in the full length construct and in the sub segments.

Figure R4, the unfolding force histogram of R3 (upper panel) and R78 (lower panel) in full-length talin rod constructs at 3.8 pN/s.

In addition, domains within 10% of a boundary can be assigned to either one of the adjacent groups. The assignment of such near-boundary domains should not affect the simulation of force fluctuation. We have clarified this grouping of the near-boundary domains in the revised manuscript.

}

12. Fig. 3d (and 4b): The traces suggest that in the particular case of R9 unfolding, that recurring unfolding and refolding increases the stability of R9, i.e. the unfolding force. While

this is likely a coincidence, could there be any reason to believe that some of the helix bundles exhibit some kind of mechanical memory?

Response:

{

We did not observe a strong correlation between number of unfolding cycles performed and unfolding force. The larger unfolding force observed at later cycles in Fig. 3d and 4b were coincidence. The unfolding forces seem to be independent of history, as shown in the figure below that contains more cycles.

Figure R5, the unfolding force-extension curves of R7-R9 in Fig. 3d of the manuscript, with more unfolding cycles and color-coded by the sequence of force cycles carried out.

}

13. Fig. 5: There are several points needing clarification in regard to the kinetic models:

- Fig. 5c: Does $\ln(K)$ refer to $\ln(k)$ (i.e. $\ln(k_u)$ and $\ln(k_f)$, respectively)? The capitalized K would otherwise indicate the ratio of the two rates, i.e. the equilibrium binding constant.

Response:

{

Thanks. We have changed K to k .

}

- More importantly, the fitted curve for the folding rate is clearly not linear in the log-lin plot, which should be the case when fitting with an Arrhenius type relation. Could the authors please explain this discrepancy?

Response:

{

We would like to thank the reviewer for the observation that the force-rate relationship is nonlinear in the log-lin plot, which is a result from the highly flexible peptide chain elasticity. It causes a force-dependent folding transition distance, which leads to such nonlinearity. A force-dependent transition distance $\Delta x_f(F)$ contributes to the transition energy by $-\int_0^F \Delta x_f(F') dF'$, resulting in a force dependent transition rate of $k_f(F) = k_f^0 \exp(\int_0^F \Delta x_f(F') dF')$, which is in general nonlinear in a log-lin plot. Only under condition when $\Delta x_f(F)$ can be approximated as a force-independent constant, $k_f(F)$ becomes linear in log-lin plot. The corresponding force dependent rate, $k_f(F) = k_f^0 \exp(F \times \Delta x_f)$, is often referred to the Bell's model.

In other words, the Bell's model is a special case of the Arrhenius relation when the transition distance is approximated as a force-independent constant. This is the case of

force-induced unfolding of protein domains where both the folded state and the transition state can be approximated as rigid bodies (Chen, H. *et al.* (2015). *JACS*, 3546(1)). In contrast, in the folding transition, due to the highly flexible nature of the polypeptide of the unfolded protein domain, the folding transition distance significantly depends on the force. Therefore, the Bell's approximation no longer holds.

Please refer to the Methods section and our previous publications (Chen, H. *et al.* (2015). *JACS*, 3546(1); You, H. *et al.* (2014). *NAR*, 42(13), 8789–95; You, H. *et al.* (2015). *JACS*, 137(7), 2424–2427.) for more details.

}

In fact, it appears that the curve for R3 unfolding given in Fig. 5e has been inserted in the 5c plot! Could the authors please correct this figure and include the actual data points with the proper Arrhenius fit?

Response:

{

Correct. The refolding rate data (red circles) and Arrhenius fitting curve (red curve) of R3 in Fig. 5c are also plotted in Fig. 5e (black crosses for experimental data and black curve for the fitting) for comparison with data from other domains. This is clarified in the revised figure captions.

}

- Also, could you please discuss the deviation of the blue data points in Fig 5c from the linear fit and to what extent the approximation of fitting the data according to the Arrhenius law is still reasonable?

Response:

{

The blue data are unfolding rates fitted using the Bell's model instead of the Arrhenius relation. As explained above and clarified in the main text and in our previous publications (Chen, H. *et al.* (2015). *JACS*, 3546(1); You, H. *et al.* (2014). *NAR*, 42(13), 8789–95; You, H. *et al.* (2015). *JACS*, 137(7), 2424–2427.), unfolding transition distance of protein domains can be approximated as a force-independent constant; therefore the Bell's model is a good approximation. The small deviation can be explained as inaccuracy due to the application of the Bell's approximation at low force regime. We chose the Bell's model to describe the force-induced unfolding rate because it predicts a simple analytical expression of the loading-rate dependent unfolding force distribution that can be directly compared to our experimental data.

}

- Was the unfolding data in this case (Fig. 5b and c) really fitted according to the Bell model, even though the data arose from a constant force experiment? If this was the case, could the authors please motivate this approach?

Response:

{

Data in Fig. 5b and c were obtained for R3 under constant force measurement. R3 underwent reversible unfolding and refolding transition; therefore the transitions rates were directly obtained by analysis of the dwell times of the respective states. The force-dependent unfolding rate data were fitted with Bell's model ($k_f(F) = k_f^0 \exp(\frac{\Delta x_{uf}^{\ddagger}}{k_B T})$), and the force-dependent refolding rate data were fitted with the more general Arrhenius relation (Fig. 5e). This is clarified in the methods section of revised manuscript.

}

- Fig. 5d: No data points for refolding rates at 4 and 5 pN for construct R7-R8 are shown and possibly are also not taken into account for the fit. Is there a reason for that or also the absence of the 4pN data in Fig. S3c?

Response:

{

Fig 5d are data for the construct R9-R12. We think the reviewer actually referred to Fig. 5e. In our experimental time scale (160s of refolding), we did not observe the refolding of R7-R8 at 4 - 5 pN; therefore the rates were only determined at three forces in 1 - 3 pN. In Fig. S3c, because the number of the refolding events is zero for both 4 pN and 5 pN, the data do not contain any information of the folding rates at these forces. In addition, because the counted values are zero, the data points are completely overlapping with each other. In the revised Fig. S3c, we have changed the symbols and connecting lines for 4 pN and 5 pN for better contrast.

Because the rates were only determined at forces in 1-3 pN, only the three data points in this force range were used for fitting by the Arrhenius relation. The fitted force-dependent rate was then used to extrapolate to forces greater than 3 pN for simulations. The extrapolated folding time scale is $< 10^{-5} \text{ s}^{-1}$ at forces ~ 4 pN, suggesting an ultra-slow refolding kinetics of R7-R8 domain at forces > 4 pN. This is consistent with the fact that we could not observe refolding over our experimental time at > 4 pN forces.

}

14. Fig. S1: Were all of the shown constructs really used and mentioned for the experiments presented in the manuscript (e.g. R7-R9 Δ R8 and F3-R1)? Otherwise I would suggest to either remove constructs from the list that are not relevant for the presented results or otherwise include the data corresponding to the extra constructs.

Response:

{

We have removed the above-mentioned constructs from the revised SI. We thank the reviewer for pointing out this discrepancy in the manuscript. In the revised Fig. S1 we have deleted R7-R9 Δ R8 and F3-R1 construct.

}

Reviewers' Comments:

Reviewer #1 (Remarks to the Author)

The authors have addressed my concerns, and I look forward to seeing this work in publication.

Reviewer #3 (Remarks to the Author)

Yan et al. satisfactorily addressed all raised points in the revised manuscript and point by point response. I thus recommend the manuscript for publication.